# PhyloVAE: Unsupervised Learning of Phylogenetic Trees via Variational Autoencoders

**Tianyu Xie**[1]**, Harry Richman**[3]**, Jiansi Gao**[3]**, Frederick A. Matsen IV**[3,4]**, Cheng Zhang**[1,2,*]

[1] School of Mathematical Sciences, Peking University
[2] Center for Statistical Science, Peking University
[3] Computational Biology Program, Fred Hutchinson Cancer Research Center
[4] Howard Hughes Medical Institute

`tianyuxie@pku.edu.cn`, `{hrichman,jsigao,matsen}@fredhutch.org`,
`chengzhang@math.pku.edu.cn`

## Abstract

Learning informative representations of phylogenetic tree structures is essential for analyzing evolutionary relationships. Classical distance-based methods have been widely used to project phylogenetic trees into Euclidean space, but they are often sensitive to the choice of distance metric and may lack sufficient resolution. In this paper, we introduce *phylogenetic variational autoencoders* (PhyloVAEs), an unsupervised learning framework designed for representation learning and generative modeling of tree topologies. Leveraging an efficient encoding mechanism inspired by autoregressive tree topology generation, we develop a deep latent-variable generative model that facilitates fast, parallelized topology generation. Phylo-VAE combines this generative model with a collaborative inference model based on learnable topological features, allowing for high-resolution representations of phylogenetic tree samples. Extensive experiments demonstrate PhyloVAE's robust representation learning capabilities and fast generation of phylogenetic tree topologies.

## 1 Introduction

Phylogenetic trees are the foundational structure for describing the evolutionary processes among individuals or groups of biological entities. Reconstructing these trees based on collected biological sequences (e.g., DNA, RNA, protein) from observed species, also known as phylogenetic inference (Felsenstein, 2004), is an essential discipline of computational biology (Fitch, 1971; Felsenstein, 1981; Yang & Rannala, 1997; Ronquist et al., 2012).

Large collections of trees obtained from these approaches (e.g., posterior samples from MCMC runs (Ronquist et al., 2012)), however, are often difficult to summarize or visualize due to the discrete and non-Euclidean nature of the tree topology space[1]. Yet, the importance of being able to do so in practice has spurred substantial research in this direction. The classical approach to visualize and analyze distributions of phylogenetic trees is to calculate pairwise distances between the trees and project them into a plane using multidimensional scaling (MDS) (Amenta & Klingner, 2002; Hillis et al., 2005; Jombart et al., 2017). However, these approaches have the shortcoming that one can not map an arbitrary point in the visualization to a tree, and therefore do not form an actual visualization of the relevant tree space. Furthermore, their effectiveness heavily depends on the choice of distance metric and can sometimes exhibit counterintuitive behaviors (Kuhner & Yamato, 2015), and the visualizations can suffer from poor resolution, where distinct sets of trees overlap within the same regions (Hillis et al., 2005; Whidden & Matsen IV, 2015) (see Figure 4 in Section 5.2 for an example). Recently, several vector representation methods have been developed for tree topologies (Liu, 2021; Penn et al., 2024). However, these representations only provide raw features whose dimension scales to the tree topology size and hence may fail to deliver concise and useful information of

---

[*]Corresponding author.
[1]In phylogenetic terminology, a *tree topology* is just the discrete graph-theoretic component of the tree without additional information such as edge lengths.

tree topologies. In general, finding good representations of tree topologies that preserve essential information for downstream tasks (e.g., comparison, visualization, clustering) remains challenging.

On the other end of the spectrum lie much more recent methods to perform density estimation on sets of phylogenetic trees (Larget, 2013; Zhang & Matsen IV, 2018; Xie & Zhang, 2023). These methods are very high resolution, as can be seen by the excellent fit that they offer to empirical distributions of phylogenetic trees. However, these methods do not lend themselves to representation learning and visualization, and it is difficult to understand what they are telling us about the structure of the phylogenetic tree shape distribution.

In this paper, we introduce *phylogenetic variational autoencoders* (PhyloVAEs), which is an unsupervised learning framework that for the first time allows both representation learning and generative modeling of phylogenetic tree topologies in a satisfying and useful way. Inspired by the tree topology generating process outlined in ARTree (Xie & Zhang, 2023), we first encode a tree topology into an integer vector representing the corresponding edge decisions, through a linear-time algorithm. Based on this encoding mechanism, we develop a deep latent-variable generative model for tree topologies, together with an inference model for the posterior distribution of the latent variable given the tree topology using learnable topological features (Zhang, 2023). In this way, PhyloVAE provides a latent space representation that can be easily visualized like the previous MDS method, but at the same time, it is a probabilistic model that gives a high-resolution representation of the tree topology distribution. Although the main purpose of PhyloVAE is representation learning, we want to emphasize that it is the generative modeling objective that forces PhyloVAE to learn high-resolution representations to retain more distributional information. We summarize our main contributions as follows:

- We propose the first representation learning framework for phylogenetic tree topologies with deep models, which has more capacity to distinguish different shapes of tree topologies compared to traditional distance-based methods (see Section 5.2 for an example). Moreover, the generative nature of PhyloVAE also allows us to map an arbitrary point in the latent space to a tree topology (Figure 3), which is impossible for current methods.
- In addition to providing a high-resolution representation of the tree topologies, PhyloVAE, as a non-autoregressive model, enjoys much faster training/generation than a previous autoregressive model ARTree (Xie & Zhang, 2023), while maintaining the approximation performance.
- Extensive and practical experiments demonstrate the robust representation ability and generative modeling efficiency of PhyloVAE for phylogenetic tree topologies.

## 2 BACKGROUND

**Phylogenetic trees** The common structure for describing evolutionary history is a phylogenetic tree, which consists of a tree topology $\tau$ and the associated non-negative edge lengths on $\tau$. The tree topology $\tau$ is a bifurcating tree graph $(V, E)$, where $V$ and $E$ represent the sets of nodes and edges, respectively. Tree topologies can be either rooted or unrooted. While our focus in this paper is on unrooted tree topologies, our proposed method can easily accommodate rooted ones as well. We will refer to unrooted tree topologies as "tree topologies" unless specified otherwise. In a tree topology $\tau$, the edges in $E$ are undirected, and the nodes in $V$ can have a degree of either

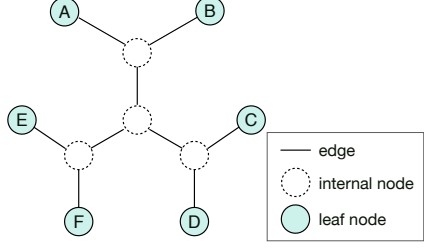

Figure 1: An example of a tree topology with six leaf nodes. The labels of leaf nodes are {A,B,C,D,E,F}.

3 or 1. Degree 3 nodes are internal nodes representing unobserved ancestor species, while degree 1 nodes are leaf nodes representing observed species labelled with corresponding species names. An edge connecting a leaf node to an internal node is called a pendant edge. The goal of phylogenetic inference is to reconstruct the underlying phylogenetic trees from the sequence data (e.g., DNA, RNA, protein) of observed species (Felsenstein, 2004), following various criteria such as maximum likelihood (Felsenstein, 1981), maximum parsimony (Fitch, 1971), and Bayesian approaches (Yang & Rannala, 1997; Mau et al., 1999; Larget & Simon, 1999; Ronquist et al., 2012). With $N$ leaf nodes, the number of possible tree topologies explodes combinatorially $((2N - 5)!!)$, posing significant challenges to phylogenetic inference and related representation learning tasks. Further details on phylogenetic models can be found in Appendix A.

**ARTree for tree topology modeling** In ARTree (Xie & Zhang, 2023), an autoregressive probabilistic model for tree topologies, a tree topology is decomposed into a sequence of leaf node addition actions, and the associated conditional probabilities are modeled using learnable topological features via graph neural networks (GNNs) (Zhang, 2023). The corresponding tree topology generating process can be described as follows. Let $\mathcal{X} = \{x_1, \ldots, x_N\}$ denote the set of leaf nodes with a pre-defined order. The generating process starts from the simplest and unique tree topology $\tau_3 = (V_3, E_3)$ that contains the first three leaf nodes $x_1, x_2, x_3$, and keeps adding new leaf node as follows. Assume an intermediate tree topology $\tau_n = (V_n, E_n)$ with the first $n < N$ leaf nodes in $\mathcal{X}$, termed an *ordinal tree topology* of rank $n$ as defined in Xie & Zhang (2023). Then, an edge $e_n \in E_n$ is selected according to a conditional distribution $Q(e_n|e_{<n})$ computed by GNNs ($e_{<n}$ is the set of previously selected edges), and $\tau_n$ is then extended to $\tau_{n+1}$ by attaching the next leaf node $x_{n+1}$ to $e_n$. This process is repeated until all leaf nodes in $\mathcal{X}$ are attached. This way, a tree topology $\tau = \tau_N$ is uniquely transformed into an edge decision sequence $D = (e_3, \ldots, e_{N-1})$, and the corresponding ARTree-based probability is $Q(\tau) = Q(D) = \prod_{n=3}^{N-1} Q(e_n|e_{<n})$ where $e_{<3} = \emptyset$. While effective, the repetitive computation of topological node embeddings and multi-round message passing steps in GNNs also add to the computational cost of ARTree. More details are deferred to Appendix B.

**Variational autoencoder** The variational autoencoder (VAE) (Kingma & Welling, 2014) assumes a generative model $p_{\boldsymbol{\theta}}(\boldsymbol{y}, \boldsymbol{z}) = p_{\boldsymbol{\theta}}(\boldsymbol{y}|\boldsymbol{z})p(\boldsymbol{z})$, where $\boldsymbol{z} \in \mathbb{R}^d$ is a latent variable with a prior distribution $p(\boldsymbol{z})$, and an inference model $q_{\boldsymbol{\phi}}(\boldsymbol{z}|\boldsymbol{y})$ as an approximation for the intractable posterior $p_{\boldsymbol{\theta}}(\boldsymbol{z}|\boldsymbol{y})$. Given an observed data set $\{\boldsymbol{y}_1, \ldots, \boldsymbol{y}_M\}$, the generative model and inference model can be jointly learned by maximizing the following evidence lower bound (ELBO)

$$L(\boldsymbol{y}; \boldsymbol{\theta}, \boldsymbol{\phi}) = \mathbb{E}_{q_{\boldsymbol{\phi}}(\boldsymbol{z}|\boldsymbol{y})} \log \left( \frac{p_{\boldsymbol{\theta}}(\boldsymbol{y}, \boldsymbol{z})}{q_{\boldsymbol{\phi}}(\boldsymbol{z}|\boldsymbol{y})} \right) = \log p_{\boldsymbol{\theta}}(\boldsymbol{y}) - D_{\mathrm{KL}} \left( q_{\boldsymbol{\phi}}(\boldsymbol{z}|\boldsymbol{y}) \| p_{\boldsymbol{\theta}}(\boldsymbol{z}|\boldsymbol{y}) \right) \leq \log p_{\boldsymbol{\theta}}(\boldsymbol{y}) \quad (1)$$

for all data points $\{\boldsymbol{y}_i : i = 1 \ldots, M\}$. Here, $p_{\boldsymbol{\theta}}(\boldsymbol{y}) = \int_{\mathbb{R}^d} p_{\boldsymbol{\theta}}(\boldsymbol{y}, \boldsymbol{z}) \mathrm{d}\boldsymbol{z}$ is the marginal likelihood of $\boldsymbol{y}$ and $D_{\mathrm{KL}}$ is the Kullback-Leibler (KL) divergence. To remedy the approximation of the latent variable posterior and achieve a more flexible generative model, the importance weighted autoencoder (IWAE) (Burda et al., 2016) instead uses the multi-sample lower bound

$$L_K(\boldsymbol{y}; \boldsymbol{\theta}, \boldsymbol{\phi}) = \mathbb{E}_{q_{\boldsymbol{\phi}}(\boldsymbol{z}^1|\boldsymbol{y}) \cdots q_{\boldsymbol{\phi}}(\boldsymbol{z}^K|\boldsymbol{y})} \log \left( \frac{1}{K} \sum_{i=1}^{K} \frac{p_{\boldsymbol{\theta}}(\boldsymbol{y}, \boldsymbol{z}^i)}{q_{\boldsymbol{\phi}}(\boldsymbol{z}^i|\boldsymbol{y})} \right) \leq \log p_{\boldsymbol{\theta}}(\boldsymbol{y}). \quad (2)$$

The equalities in equation (1) and (2) hold if and only if $q_{\boldsymbol{\phi}}(\boldsymbol{z}|\boldsymbol{y}) = p_{\boldsymbol{\theta}}(\boldsymbol{z}|\boldsymbol{y})$. While VAEs have been effectively used in graph representation learning (Kipf & Welling, 2016; Simonovsky & Komodakis, 2018; Winter et al., 2021; Zahirnia et al., 2022), they typically require transforming graphs into numerical encodings, such as adjacency matrices, and then working in the encoding space. However, the specific bifurcating structure of phylogenetic tree topologies imposes unique constraints on these adjacency matrices, posing challenges for applying these methods directly.

## 3 PROPOSED METHOD

In this section, we introduce phylogenetic variational autoencoders (PhyloVAEs), an unsupervised learning framework designed specifically for phylogenetic tree topologies. We begin with a concise overview of the problem setting and the fundamental components of PhyloVAE in Section 3.1. We then develop an encoding mechanism that bijectively maps tree topologies to encoding vectors in Section 3.2. Finally, Section 3.3 elucidates how to utilize this encoding mechanism and learnable topological features to establish the generative and inference models within the PhyloVAE framework. We emphasize that the input to our algorithms is a collection of phylogenetic tree topologies [2]. Our goal is to build a probabilistic model and to learn useful representations of this collection, not to infer those trees directly from sequence data, which is a separate and intensively studied problem.

---

[2]These collections often come from phylogenetic analysis softwares such as MrBayes (Ronquist et al., 2012) and BEAST (Drummond & Rambaut, 2007), but they can also come from observations, open-source data sets, and other biological softwares, as long as they are of scientific interest.

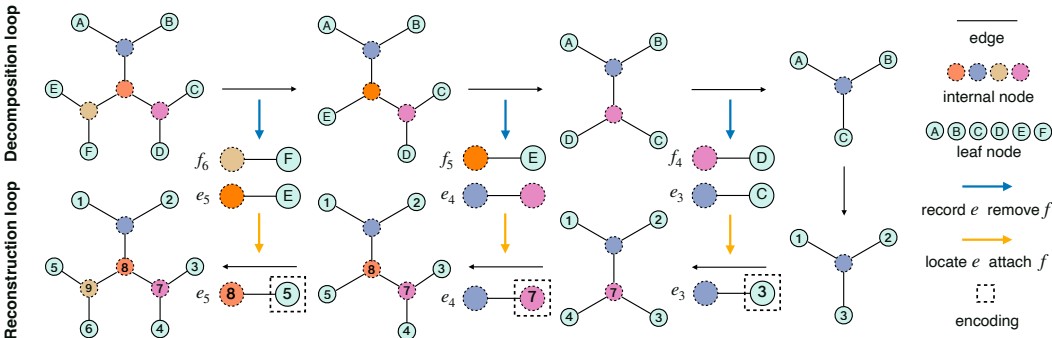

Figure 2: The decomposition loop and reconstruction loop for encoding the tree topology with leaf nodes $\mathcal{X} = \{A,B,C,D,E,F\}$ in Figure 1. Starting from the tree topology in the upper left, we remove the pendant edges $f_6, f_5, f_4$ (associated with the leaf nodes F, E, and D) sequentially, and record edge decision $e_5, e_4, e_3$. Then starting from the three-leaf tree topology in the lower right, we add back $f_4, f_5, f_6$ and index these nodes (except for the root) sequentially. The resulting encoding vector is $(3, 7, 5)$, which are the indexes associated with $e_3, e_4, e_5$.

## 3.1 PHYLOVAE: AN OVERVIEW

Let $\mathcal{T} = \{(\tau^i, w^i)\}_{i=1}^M$ be a collection of tree topologies[3], where $w^i$ is the weight for the tree topology $\tau^i$ and $\sum_{i=1}^M w^i = 1$. All tree topologies in $\mathcal{T}$ have the same leaf nodes $\mathcal{X} = \{x_1, x_2, \ldots, x_N\}$ with a pre-selected order. For example, $\mathcal{T}$ can be a sample of tree topologies produced by some phylogenetic inference software, such as from a posterior sample (Ronquist et al., 2012; Suchard et al., 2018) or the bootstrap (Felsenstein, 1985; Minh et al., 2013), where the weight can equal to the frequency of a tree topology among these samples. Given the observed data set $\mathcal{T}$, the data distribution is defined as $p_{\text{data}}(\tau) = \sum_{i=1}^M w^i \delta_{\tau^i}(\tau)$ where $\delta$ is a Kronecker delta function that satisfies $\delta_{\tau^i}(\tau) = 1$ if $\tau = \tau^i$ and $\delta_{\tau^i}(\tau) = 0$ elsewhere.

Similar to standard VAEs, PhyloVAE consists of a generative model and an inference model. Let $\boldsymbol{z} \in \mathbb{R}^d$ be a latent variable with a prior distribution $p(\boldsymbol{z})$ and $p_{\boldsymbol{\theta}}(\tau|\boldsymbol{z})$ be a probabilistic model that defines the probability of generating $\tau$ conditioned on the latent variable $\boldsymbol{z}$. The marginal probability of $\tau$ is given by $p_{\boldsymbol{\theta}}(\tau) = \int_{\mathbb{R}^d} p_{\boldsymbol{\theta}}(\tau|\boldsymbol{z})p(\boldsymbol{z})\mathrm{d}\boldsymbol{z}$. The prior distribution $p(\boldsymbol{z})$ is required to be analytic and easy to sample from. In this paper, we will use a standard Gaussian prior distribution, i.e., $p(\boldsymbol{z}) = \mathcal{N}(\boldsymbol{z}; \boldsymbol{0}_d, \boldsymbol{I}_d)$. With an inference model $q_{\boldsymbol{\phi}}(\boldsymbol{z}|\tau)$ that approximates the posterior $p_{\boldsymbol{\theta}}(\boldsymbol{z}|\tau)$, the multi-sample lower bound on $\tau$ takes the form

$$L_K(\tau; \boldsymbol{\theta}, \boldsymbol{\phi}) = \mathbb{E}_{q_{\boldsymbol{\phi}}(\boldsymbol{z}^1|\tau)\cdots q_{\boldsymbol{\phi}}(\boldsymbol{z}^K|\tau)} \log\left(\frac{1}{K}\sum_{i=1}^K \frac{p_{\boldsymbol{\theta}}(\tau, \boldsymbol{z}^i)}{q_{\boldsymbol{\phi}}(\boldsymbol{z}^i|\tau)}\right) \leq \log p_{\boldsymbol{\theta}}(\tau), \qquad (3)$$

which reduces to the standard ELBO for $\log p_{\boldsymbol{\theta}}(\tau)$ when $K = 1$. The overall multi-sample lower bound on $\mathcal{T}$ is defined as $L_K(\mathcal{T}; \boldsymbol{\theta}, \boldsymbol{\phi}) = \mathbb{E}_{p_{\text{data}}(\tau)} L_K(\tau; \boldsymbol{\theta}, \boldsymbol{\phi})$, which serves as the objective function for training PhyloVAE. Unlike standard VAEs, the discrete nature of $\tau$ makes it challenging to construct the generative model and inference model. In what follows, we describe how this is done using an encoding mechanism and learnable topological features respectively.

## 3.2 AN ENCODING MECHANISM FOR TREE TOPOLOGIES

Recall that in the generating process of ARTree, there exists a map between a tree topology $\tau$ and the edge decision sequence $D = (e_3, \ldots, e_{n-1})$ (see Section 2 and Appendix B for more details). We can further encode this sequence into an integer-valued vector via the following procedure.

**Decomposition loop** In this loop, we repetitively remove the leaf nodes from $\tau_N = \tau$, an ordinal tree topology of rank $N$, starting from the last added node $x_N$ to the first added node $x_4$. Suppose now we have $\tau_n = (V_n, E_n)$ in hand, a pendant edge $f_n = (t_n, x_n)$ is firstly removed, where $t_n$ is

---

[3]The weights are formed by merging replicate tree topologies in the data set for a compact form of $\mathcal{T}$.

---

**Algorithm 1:** A linear-time algorithm for tree topology encoding

---

**Input:** A tree topology $\tau$ with $N$ leaf nodes.
**Output:** An encoding vector $\boldsymbol{s}(\tau) = (s_3, \ldots, s_{N-1}) \in \mathbb{N}^{N-3}$ corresponding to $\tau$.
$\tau_N \leftarrow \tau$;
**for** $n = N, \ldots, 4$ **do**
   | Remove the pendant edge $f_n = (t_n, x_n)$ from $\tau_n$ and obtain $\tau_{n-1}$;
   | Record the edge decision $e_{n-1} = (u_{n-1}, v_{n-1})$ on $\tau_{n-1}$;
**end**
$\text{Index}(x_1) \leftarrow 1; \text{Index}(x_2) \leftarrow 2; \text{Index}(x_3) \leftarrow 3; r \leftarrow$ the unique internal node of $\tau_3$;
**for** $n = 3, \ldots, N-1$ **do**
   | Attach the pendant edge $f_{n+1}$ to $e_n$ (assume $e_n = (u_n, v_n)$ is directed away from $r$) and
      obtain $\tau_{n+1}$;
   | $\text{Index}(t_{n+1}) \leftarrow N + n - 2; \text{Index}(x_{n+1}) \leftarrow n + 1$;
   | $s_n \leftarrow \text{Index}(v_n)$;
**end**

---

the internal node connected to $x_n$, resulting in an ordinal tree topology of rank $n - 1$, $\tau_{n-1}$. Let the two neighbors of $t_n$ in $\tau_n$ be $u_{n-1}$ and $v_{n-1}$ (in addition to $x_n$). The edge $e_{n-1} = (u_{n-1}, v_{n-1})$, therefore, is the corresponding edge decision for $x_n$ on $\tau_{n-1}$ which we save for the reconstruction loop. This procedure continues until a tree topology $\tau_3$ with the first three leaf nodes is reached. See the top row in Figure 2 for an illustration.

**Reconstruction loop** In this loop, we start from $\tau_3$ and reconstruct $\tau_N$ by adding back the leaf nodes from $x_4$ to $x_N$. At the beginning, we index the leaf nodes on $\tau_3$ by setting $\text{Index}(x_1) = 1, \text{Index}(x_2) = 2, \text{Index}(x_3) = 3$, and denote the unique internal node in $\tau_3$ as $r$. Suppose now we have an ordinal tree topology $\tau_n = (V_n, E_n)$ of rank $n$. We first locate the edge decision $e_n = (u_n, v_n)$ on $\tau_n$ (without loss of generality, $e_n$ is assumed to be directed away from $r$ so that it can be identified via $v_n$ on $\tau_n$) and then attach the pendant edge $f_{n+1} = (t_{n+1}, x_{n+1})$ to $e_n$. Finally, we set $\text{Index}(t_{n+1}) = N + n - 2$ and $\text{Index}(x_{n+1}) = n + 1$. In this way, the next tree topology $\tau_{n+1} = (V_{n+1}, E_{n+1})$ is constructed and all the nodes except $r$ in $V_{n+1}$ are indexed. This process ends upon the full reconstruction of $\tau_N$. See the bottom row in Figure 2 for an illustration.

After the above two loops, the encoding vector $\boldsymbol{s}(\tau)$ for a tree topology $\tau$ takes the form

$$\boldsymbol{s}(\tau) = [s_3, s_4, \cdots, s_{n-1}]' = [\text{Index}(v_3), \text{Index}(v_4), \cdots, \text{Index}(v_{n-1})]' \in \mathbb{N}^{n-3}. \quad (4)$$

This encoding mechanism is summarized in Algorithm 1. Theorem 1 (proof in Appendix C.1) shows that this mechanism has linear time complexity, which is crucial to the efficient training of PhyloVAE.

**Theorem 1.** *Given a tree topology $\tau$ with $N$ leaf nodes, the time complexity of computing its encoding vector $\boldsymbol{s}(\tau)$ is $O(N)$.*

Conversely, when given an encoding vector $\boldsymbol{s}$, one can simply follow the reconstruction loop to obtain the corresponding tree topology $\tau$ in linear time (deduced from Theorem 1). This enables the fast generation of samples from PhyloVAE (details are deferred to Appendix C.2). We note that a similar encoding strategy has been proposed in Phylo2Vec (Penn et al., 2024). However, their approach has quadratic time complexity for vector encoding of tree topologies with unlabelled internal nodes. Our method achieves faster processing by employing a smart indexing strategy, eliminating the need for repetitive relabelling of edges during the reconstruction loop.

### 3.3 GENERATIVE MODEL AND INFERENCE MODEL IN PHYLOVAE

**Generative model** The encoding mechanism in Section 3.2 allows us to define $p_{\boldsymbol{\theta}}(\tau|\boldsymbol{z})$ through $p_{\boldsymbol{\theta}}(\boldsymbol{s}(\tau)|\boldsymbol{z})$, where $\boldsymbol{s}(\tau) = (s_3, \ldots, s_{N-1})$ is the encoding vector for $\tau$. Similar to the diagonal Gaussian distribution used in standard VAEs, we assume the elements in $\boldsymbol{s}(\tau)$ are conditionally independent given $\boldsymbol{z}$, i.e.,

$$p_{\boldsymbol{\theta}}(\tau|\boldsymbol{z}) = p_{\boldsymbol{\theta}}(\boldsymbol{s}(\tau)|\boldsymbol{z}) = \prod_{n=3}^{N-1} p_{\boldsymbol{\theta}}(s_n|\boldsymbol{z}). \quad (5)$$

The factorized form of equation (5) enables parallel computation of the probabilities $p_{\boldsymbol{\theta}}(s_n|\boldsymbol{z})$. Our experiments show that this non-autoregressive structure substantially reduces the computational cost compared to autoregressive models such as ARTree.

Let $S_n := \{i \in \mathbb{N} : 1 \leq i \leq n \text{ or } N+1 \leq i \leq N+n-3\}$ be the set of allowable indices for the edges $\{\text{Index}(v_n) : e_n = (u_n, v_n) \in E_n\}$. For all $3 \leq n < N$, $p_{\boldsymbol{\theta}}(s_n|\boldsymbol{z})$ takes the form

$$p_{\boldsymbol{\theta}}(s_n|\boldsymbol{z}) = \text{Discrete}\left[\text{softmax}(\boldsymbol{m}_n \odot \boldsymbol{\phi}_n(\boldsymbol{z}))\right], \tag{6a}$$

$$\boldsymbol{\Phi}(\boldsymbol{z}) = [\boldsymbol{\phi}_3(\boldsymbol{z}), \ldots, \boldsymbol{\phi}_{N-1}(\boldsymbol{z})]' = \text{MLP}_{\text{enc}}(\boldsymbol{z}) \in \mathbb{R}^{(N-3)\times(2N-3)}, \tag{6b}$$

where $\text{MLP}_{\text{enc}}$ is a multi-layer perceptron, $\odot$ is elementwise multiplication, and $\boldsymbol{m}_n$ is defined as

$$(\boldsymbol{m}_n \odot \boldsymbol{\alpha})_i = \left\{ \begin{array}{ll} \boldsymbol{\alpha}_i, & i \in S_n, \\ -\infty, & \text{elsewhere.} \end{array} \right. \tag{7}$$

This mask vector is introduced to ensure that the generated encoding vector $\boldsymbol{s}$ is always valid for representing a tree topology.

**Inference model**  The inference model $q_{\boldsymbol{\phi}}(\boldsymbol{z}|\tau)$ is built on top of learnable topological features as follows. Firstly, we compute the topological node embeddings $\{\boldsymbol{f}_u^0 : u \in V\}$ for $\tau = (V, E)$ by minimizing the following Dirichlet energy $\mathcal{E}(\tau) = \sum_{(u,v)\in E} \|\boldsymbol{f}_u^0 - \boldsymbol{f}_v^0\|^2$ using the efficient two-pass algorithm described in Zhang (2023). These topological node embeddings are then fed into a gated message-passing block (Li et al., 2015) implemented as

$$\boldsymbol{f}_u^{l+1} = \text{GRU}\left(\boldsymbol{f}_u^l, \sum_{v\in\mathcal{N}(u)} \boldsymbol{W}_{\text{msg}}^l \boldsymbol{f}_v^l\right), \ u \in V, \tag{8}$$

where $\mathcal{N}(u)$ is the neighborhood of $u$, $\boldsymbol{W}_{\text{msg}}^l$ is a learnable message matrix that aggregate the information from $N(u)$, and GRU is a gated recurrent unit (Cho et al., 2014). After $L$ rounds of message passing, the graph embedding vector $\boldsymbol{f}_\tau$ is computed by a sum-pooling layer, i.e., $\boldsymbol{f}_\tau = \sum_{u\in V} \boldsymbol{f}_u^L$. Finally, we use a diagonal normal distribution for the conditional distribution of the latent variable $\boldsymbol{z}$ whose mean and standard deviation are defined based on $\boldsymbol{f}_\tau$ as follows

$$q_{\boldsymbol{\phi}}(\boldsymbol{z}|\tau) = \mathcal{N}\left(\boldsymbol{z}; \boldsymbol{\mu}_\tau, \text{diag}(\boldsymbol{\sigma}_\tau^2)\right), \ \boldsymbol{\mu}_\tau = \text{MLP}_\mu(\boldsymbol{f}_\tau), \ \log\boldsymbol{\sigma}_\tau = \text{MLP}_\sigma(\boldsymbol{f}_\tau), \tag{9}$$

where $\text{MLP}_\mu$ and $\text{MLP}_\sigma$ are two multi-layer perceptrons, and $\boldsymbol{\phi}$ are the learnable parameters. The mean of the inference model, $\boldsymbol{\mu}_\tau \in \mathbb{R}^d$, is a deterministic low-dimensional representation of $\tau$, and we will show its representation power in our experiments. Although this inference model is built on top of Zhang (2023), Zhang (2023) only provides a deep model architecture for extracting graph features and does not use it for representation learning. This representation also induces a pairwise distance between tree topologies. For two tree topologies $\tau_1, \tau_2$, we define the $L^p$ distance between them as $D_{L^p}(\tau_1, \tau_2) = \|\boldsymbol{\mu}_{\tau_1} - \boldsymbol{\mu}_{\tau_2}\|_p$, where $\|\cdot\|_p$ is the $p$-norm in the Euclidean space. Note that the generative model also allows us to map an arbitrary point in the latent space to a tree topology, which is impossible for previous representation methods.

**Optimization**  Thanks to the Gaussian inference model, the gradient $\nabla_{\boldsymbol{\theta},\boldsymbol{\phi}} L_K(\tau; \boldsymbol{\theta}, \boldsymbol{\phi})$ can be derived using the reparameterization trick (Kingma & Welling, 2014) as follows

$$\nabla_{\boldsymbol{\theta},\boldsymbol{\phi}} L_K(\tau; \boldsymbol{\theta}, \boldsymbol{\phi}) = \mathbb{E}_{\boldsymbol{\varepsilon}^1,\ldots,\boldsymbol{\varepsilon}^K \sim \mathcal{N}(\cdot;\boldsymbol{0},\boldsymbol{I})} \nabla_{\boldsymbol{\theta},\boldsymbol{\phi}} \log\left(\frac{1}{K}\sum_{i=1}^{K} \frac{p_{\boldsymbol{\theta}}(\tau, \boldsymbol{\mu}_\tau + \boldsymbol{\sigma}_\tau \odot \boldsymbol{\varepsilon}^i)}{q_{\boldsymbol{\phi}}(\boldsymbol{\mu}_\tau + \boldsymbol{\sigma}_\tau \odot \boldsymbol{\varepsilon}^i|\tau)}\right). \tag{10}$$

During training, parameters of the generative model and inference model are updated along the gradient direction $\nabla_{\boldsymbol{\theta},\boldsymbol{\phi}} L_K(\mathcal{T}; \boldsymbol{\theta}, \boldsymbol{\phi}) = \mathbb{E}_{p_{\text{data}}(\tau)} \nabla_{\boldsymbol{\theta},\boldsymbol{\phi}} L_K(\tau; \boldsymbol{\theta}, \boldsymbol{\phi})$, using Monte Carlo gradient estimators.

## 4 RELATED WORKS

For harnessing the latent-variable structure to accelerate autoregressive models, Gu et al. (2018) proposed non-autoregressive machine translation by defining a factorizable distribution for the output sequence conditioned on the input sequence and latent fertility variable. Kaiser et al. (2018) extended

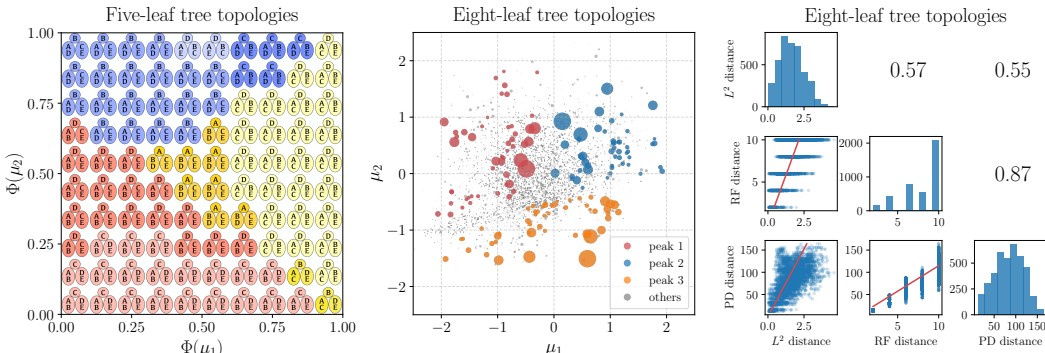

Figure 3: Performance of PhyloVAE for structural representation on simulated data sets. **Left**: A visualization of the 2D latent manifold for the data set of five-leaf tree topologies. $\Phi(\cdot)$ refers to the cumulative density function of the one-dimensional standard Gaussian distribution. Different colors represent the first edge decision and different transparencies of each color represent the second edge decision. **Middle**: Representation vectors of all the eight-leaf tree topologies. The scatter size is proportional to the probability of the corresponding tree topology. **Right**: Pairwise scatter plot, linear regression, and Pearson correlation coefficients between different distance metrics of tree topologies. $L^2$ = Euclidean distance in PhyloVAE latent space, RF = Robinson-Foulds, PD = Path difference.

this to discrete latent variables that summarize the input information. This approach was also integrated with normalizing flows by Ma et al. (2019).

Previous VAE frameworks for graph representation learning often encode a graph to its adjacency matrix and then define the generative models for matrices (Kipf & Welling, 2016; Simonovsky & Komodakis, 2018; Winter et al., 2021; Zahirnia et al., 2022). However, the bifurcating structure and the unlabelled internal nodes of tree topologies put special constraints on adjacency matrices, which may hinder the application of previous works to representation learning of phylogenetic trees.

The most popular means of learning an embedding of a collection of phylogenetic trees is to calculate pairwise distances in some way and project to a Euclidean space using multidimensional scaling (Amenta & Klingner, 2002; Hillis et al., 2005; Jombart et al., 2017). More recently, Penn et al. (2024) proposed an encoding strategy that relies on tree topology branching patterns, with representation dimensions scaling according to tree size.

Some previous works integrated trees with VAEs (Shin et al., 2017; Vikram et al., 2018; Manduchi et al., 2023). However, they all consider a tree-shaped prior distribution or hierarchical latent variable structure for enhanced interpretability and generative quality. These papers do not consider modeling any graph or tree objects and thus are clearly distinct from our PhyloVAE.

## 5 EXPERIMENTS

In this section, we evaluate the performance of PhyloVAE for structural representation on simulated data sets (Section 5.1) and real phylogenies (Section 5.2), and generative modeling on benchmark data sets (Section 5.3). We set the latent dimension $d = 2$ for better visualization of the latent representations in Section 5.1 and 5.2. For all experiments, the number of particles is set to $K = 32$ and the inference model employs $L = 2$ rounds of message passing. Leaf nodes are ordered lexicographically based on the corresponding species names. We designed our experiments with the goals of (i) verifying the effectiveness of PhyloVAE for representation learning of tree topologies and (ii) examining the generative modeling performance of PhyloVAE, with an emphasis on the generation speed. Results are gathered after 200,000 iterations with a batch size of 10. Further details can be found in Appendix D. Our code is released at `https://github.com/tyuxie/PhyloVAE`.

### 5.1 REPRESENTATION LEARNING ON SIMULATED DATA SETS

**Five-leaf tree topologies** In this experiment, the training set consists of all the 15 tree topologies with five leaves and a randomly generated weight vector $\boldsymbol{w} \sim \text{Dirichlet}(\beta\mathbf{1}) \in \mathbb{R}^{15}$ for these tree

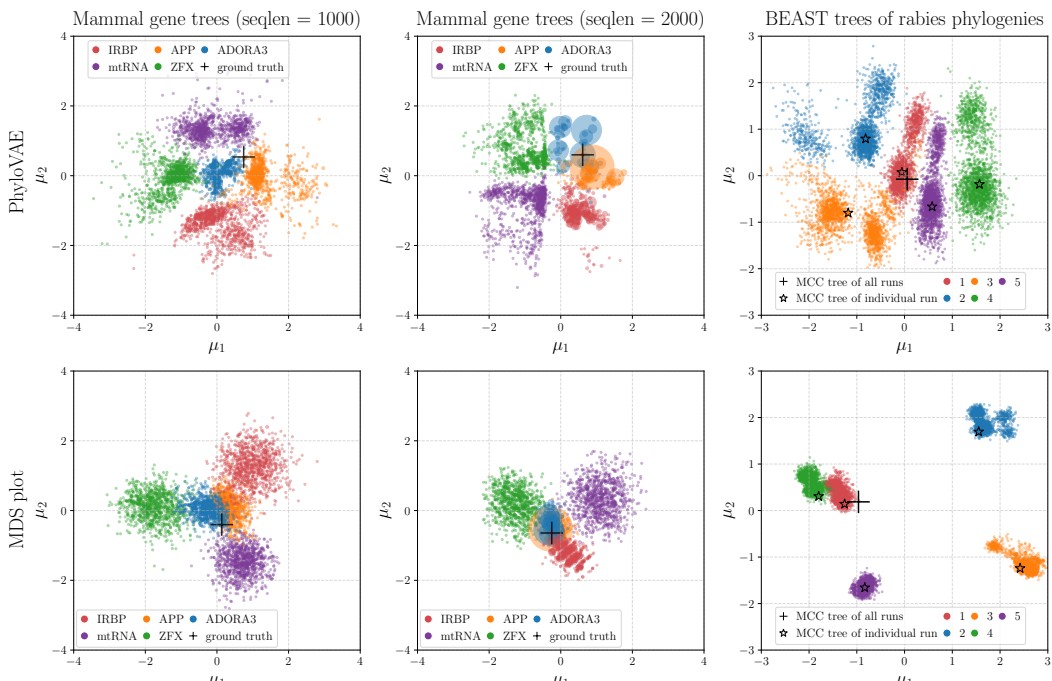

Figure 4: Performances of PhyloVAE and MDS plot for representing real phylogenies. **Left/Middle**: Latent representations of the posterior mammal gene trees for five genes with different lengths. The scatter size is proportional to the probability of the tree topology. **Right**: Latent representations of samples of tree topologies from five independent BEAST runs on the rabies data set.

topologies with $\beta = 0.75$. Figure 3 (left) depicts the tree topologies as partition representation (see Figure 7 in Appendix D.1 for an illustration), generated by a trained PhyloVAE conditioned on the Gaussian quantiles. To ease presentation, we choose the argmax index of the multinomial probability in equation (6a) instead of randomly sampling. We see that the generated tree topology exhibits nice continuity as the latent variable varies.

**Eight-leaf tree topologies**    We begin by constructing the training set, which consists of a mixture of three peaked distributions comprising all 10,395 tree topologies with eight leaves. Each peaked distribution is derived as follows: we select a ground truth tree topology and simulate DNA sequences of length 500 for the eight leaf nodes using the Jukes-Cantor (JC) substitution model (Jukes et al., 1969) with the edge lengths fixed at 1. We then compute the posterior distribution of tree topologies given the simulated DNA sequences with a uniform prior over tree topologies and the edge lengths fixed at 1. This posterior distribution is used as the peaked distribution as it concentrates around the selected ground truth tree topology. The same procedure is repeated three times (see Figure 8(a), 8(b) and 8(c) for the selected ground truth tree topologies), leading to three peaked distributions which are then evenly mixed to form the training data set.

The latent representations of eight-leaf tree topologies are visualized in Figure 3 (middle), where tree topologies on different peaks are clearly separated, demonstrating the effectiveness of PhyloVAE for representation learning. Following Kendall & Colijn (2016), we compare $L^2$ distance to Robinson-Foulds (RF) distance (Robinson & Foulds, 1981) and path difference (PD) distance (Steel & Penny, 1993) in Figure 3 (right), where $L^2$ distance shows a positive correlation with RF/PD distance.

## 5.2    REPRESENTATION LEARNING ON REAL PHYLOGENIES

**Gene trees and sequence lengths**    In this experiment, PhyloVAE is employed to analyze phylogenetic inference results obtained with different genes (different genes evolve under different evolutionary models) and sequence lengths. Following Hillis et al. (2005), we select five genes and the ground truth phylogenetic tree (Figure 9; 44 leaves) from the early placental mammal evolution analysis in Murphy et al. (2001). For each gene, we simulate the DNA sequences with a fixed

Table 1: KL divergences to the ground truth of different methods across eight benchmark data sets. "PhyloVAE ($d$)" means PhyloVAE with latent dimension $d$. The "# Training set" and "# Ground truth" columns show the number of unique tree topologies in the training sets and ground truth respectively. The results are averaged over 10 replicate training sets. The tree topology probability of PhyloVAE is estimated using importance sampling with 1000 particles. The results of SBN-EM, SBN-EM-$\alpha$ are from Zhang & Matsen IV (2018), and the results of ARTree are from Xie & Zhang (2023). For each data set, the best result is marked in **black bold font** and the second best result is marked in **brown bold font**.

| Sequence set | # Leaves | # Training set | # Ground truth | KL divergence to ground truth | | | | |
| --- | --- | --- | --- | --- | --- | --- | --- | --- |
| | | | | SBN-EM | SBN-EM-$\alpha$ | ARTree | PhyloVAE (2) | PhyloVAE (10) |
| DS1 | 27 | 1228 | 2784 | 0.0136 | **0.0130** | **0.0045** | 0.0273 | 0.0189 |
| DS2 | 29 | 7 | 42 | 0.0199 | 0.0128 | **0.0097** | 0.0100 | **0.0098** |
| DS3 | 36 | 43 | 351 | 0.1243 | 0.0882 | 0.0548 | **0.0529** | **0.0477** |
| DS4 | 41 | 828 | 11505 | 0.0763 | 0.0637 | **0.0299** | 0.0619 | **0.0469** |
| DS5 | 50 | 33752 | 1516877 | 0.8599 | 0.8218 | **0.6266** | 0.7985 | **0.5744** |
| DS6 | 50 | 35407 | 809765 | 0.3016 | 0.2786 | **0.2360** | 0.3241 | **0.2207** |
| DS7 | 59 | 1125 | 11525 | 0.0483 | 0.0399 | **0.0191** | 0.0591 | **0.0370** |
| DS8 | 64 | 3067 | 82162 | 0.1415 | 0.1236 | **0.0741** | 0.1372 | **0.1061** |

length along the ground truth tree using the corresponding evolutionary model, run a MrBayes chain (Ronquist et al., 2012) for one million iterations, and sample per 100 iterations in the last 100,000 iterations, to gather the posterior samples, as done in Hillis et al. (2005). These one million iterations are enough for the MrBayes run to converge. These 5,000 tree topologies with uniform weights constitute the training set of PhyloVAE.

The upper left and upper middle plots of Figure 4 depict the learned latent representations of the posterior tree topologies for different genes obtained by PhyloVAE. We see that different groups are clearly separated. With a sequence length of 1000, the inferred posteriors from ADORA3 and APP are close to the ground truth tree topology, while those from IRBP, mtRNA, and ZFX show deviations. When the sequence length is increased to 2000, the inferred posterior from ADORA3 becomes more concentrated around the ground truth tree topology, while the inferred posteriors from the other genes remain diffuse. Figure 10 in Appendix E.1 shows that all five genes discover the ground truth tree topology with a sequence length of 5000. For baseline visualization methods, the lower left and lower middle plots of Figure 4 show the multidimensional scaling (MDS) plots (Hamer & Young, 2013) of mammal gene trees, where we find that different groups tend to concentrate towards the origins and overlap with each other, while our PhyloVAE provides more clear representations of different groups.

**Multiple phylogenetic analyses comparison** In this experiment, we use PhyloVAE to compare multiple phylogenetic analyses and assess convergence. The sequence alignment under consideration comprises 290 rabies genomes (Viana et al., 2023). We conduct 5 independent BEAST (Suchard et al., 2018) analyses for 400 million iterations, and sample every 100,000 iterations in the last 200 million iterations. Afterward, the rooted posterior tree topologies sampled by BEAST are unrooted. The resulting 10,000 tree topologies with uniform weights form the training set of PhyloVAE.

In Figure 4 (right), tree topologies from five independent BEAST runs form five non-overlapping groups, with 2-3 sub-groups within each group, indicating the divergence of these BEAST runs. Notably, the maximum clade credibility (MCC) tree from each run resides within the correct high-density region in the latent space. An example showcasing the convergence of independent phylogenetic analyses is provided in Appendix E.2.

## 5.3 GENERATIVE MODELING ON BENCHMARK DATA SETS

Finally, we assess the generative modeling performance of PhyloVAE on eight benchmark sequence sets, DS1-8, which contain biological sequences from 27 to 64 eukaryote species and are commonly considered for benchmarking tree topology density estimation and Bayesian phylogenetic inference tasks in previous works (Zhang & Matsen IV, 2018; 2019; 2024; Zhang, 2020; Mimori & Hamada, 2023; Zhou et al., 2023; Xie & Zhang, 2023; Xie et al., 2024a;b; Molén et al., 2024; Hotti et al., 2024). These eight data sets cover comprehensive posterior patterns (Whidden & Matsen IV, 2015) and are considered good representative cases. The training sets and ground truths for PhyloVAE are obtained the same way as in Zhang & Matsen IV (2018) (see more details in Appendix D.5). We consider SBN-EM, SBN-EM-$\alpha$, and ARTree as baselines, and use the same experimental settings in

the original papers (Zhang & Matsen IV, 2018; Xie & Zhang, 2023). For a fair comparison, we uses the same batch size and number of iterations for PhyloVAE as in ARTree.

Table 1 reports the approximation accuracies measured by KL divergence obtained by different methods. We see that PhyloVAE with a latent dimension of 10 achieves comparable (if not better) results to ARTree. Although PhyloVAE with a latent dimension of 2 may have reduced capacity, we find it works fairly well in DS1-8, validating the reliability of the two-dimensional representations in Sections 5.1 and 5.2. Figure 5 shows the runtime comparison between ARTree and PhyloVAE ($d = 10$) with $K = 32$ particles. We see that both the training time and generation time of PhyloVAE are significantly reduced compared to ARTree, even though multiple particles are used in the multi-sample lower bound (3). This is due to the non-autoregressive nature of PhyloVAEs, as inherited from VAEs. Note that PhyloVAE achieves these while providing useful high-resolution representations of phylogenetic tree topologies (see Figure 14 in Appendix E.3), which is impossible for the other baseline methods that are merely designed for tree density estimation.

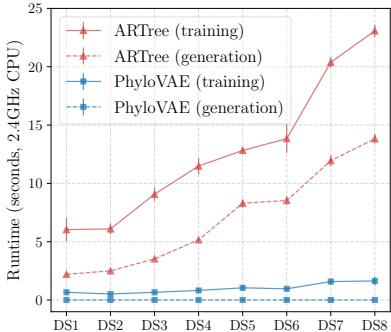

Figure 5: Runtime comparison between ARTree and PhyloVAE ($d = 10$) with $K = 32$ particles. Training means 10 training iterations. Generation means generating 100 tree topologies.

## 6 CONCLUSION

This paper proposes PhyloVAE, an unsupervised learning framework designed for representation learning and generative modeling of phylogenetic tree topologies. By incorporating an efficient encoding mechanism of tree topologies and leveraging a latent-variable architecture, PhyloVAE facilitates fast training and generation using non-autoregressive generative models. The learned inference model also provides informative, high-resolution representations for tree topologies, as demonstrated in our experiments. PhyloVAE thus offers a latent space representation that can be easily visualized like the previous MDS method, while also functioning as a probabilistic model that provides a high-resolution view of tree topology distributions. Further applications of PhyloVAE to practical tasks in phylogenetics (e.g., phylogenetic placement, Bayesian phylogenetic inference, etc.) and extending PhyloVAE for tree topologies with different leaves would be interesting future directions.

## ACKNOWLEDGEMENTS

This work was supported by National Natural Science Foundation of China (grant no. 12201014, grant no. 12292980 and grant no. 12292983), as well as National Institutes of Health grant AI162611. The research of Cheng Zhang was support in part by National Engineering Laboratory for Big Data Analysis and Applications, the Key Laboratory of Mathematics and Its Applications (LMAM) and the Key Laboratory of Mathematical Economics and Quantitative Finance (LMEQF) of Peking University. Frederick Matsen is an investigator of the Howard Hughes Medical Institute. The authors appreciate Dave Rich for carefully validating the codebase. The authors are grateful for the computational resources provided by the High-performance Computing Platform of Peking University. The authors appreciate the anonymous ICLR reviewers for their constructive feedback.

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

## A  PHYLOGENETIC TREES AND PHYLOGENETIC INFERENCE

The common structure for describing evolutionary history is a phylogenetic tree, which consists of a bifurcating tree topology $\tau$ and the associated non-negative edge lengths $\boldsymbol{q}$ (see Figure 9 for a real example). The tree topology $\tau$ is a bifurcating tree graph $(V, E)$, where $V$ and $E$ are the set of nodes and edges respectively. We will be consistent with the main text and use "tree topology" for unrooted tree topology. For a tree topology $\tau$, the edges in $E$ are undirected and the nodes in $V$ can have either 3 degrees or 1 degree. Those with 3 degrees are called internal nodes, representing unobserved ancestor species without labels, and those with 1 degree are called leaf nodes, representing the observed existing species with labels corresponding to the species names. An edge is called a pendant edge if it connects one leaf node to an internal node. For each edge $e \in E$, there exists a corresponding non-negative edge length $q_e$, which quantifies the evolutionary intensity between two neighboring species. The set of edge lengths on $\tau$ is given by $\boldsymbol{q} = \{q_e : e \in E\}$.

Each leaf node in $V$ corresponds to a species with an observed biological sequence (e.g., DNA, RNA, protein). Let $\boldsymbol{Y} = \{Y_1, \ldots, Y_M\} \in \Omega^{N \times M}$ denote the observed sequences (with characters in $\Omega$) of $M$ sites across $N$ species. The transition probabilities of the characters along the edges of a phylogenetic tree are commonly modeled using a continuous-time Markov chain (Felsenstein, 2004). With the common assumption that different sites evolve independently and identically, given a phylogenetic tree $(\tau, \boldsymbol{q})$, the likelihood of observing $\boldsymbol{Y}$ is

$$p(\boldsymbol{Y}|\tau, \boldsymbol{q}) = \prod_{i=1}^{M} \sum_{a^i} \eta(a_r^i) \prod_{(u,v) \in E} P_{a_u^i a_v^i}(q_{uv}), \tag{11}$$

where $a^i$ ranges over all extensions of $Y_i$ to the internal nodes with $a_u^i$ being the character assignment of node $u$ ($r$ represents an arbitrary internal node as the virtual root node), $E$ is the set of edges of $\tau$, $q_{uv}$ is the branch length of the edge $(u, v) \in E$, $P_{jk}(q)$ is the transition probability from character $j$ to $k$ through an edge of length $q$, and $\eta$ is the stationary distribution of the Markov chain. Parameters in equation (11) are called an *evolutionary model*. Assuming a prior distribution $p(\tau, \boldsymbol{q})$ on phylogenetic trees, Bayesian phylogenetic inference aims at properly estimating the posterior distribution

$$p(\tau, \boldsymbol{q}|\boldsymbol{Y}) = \frac{p(\boldsymbol{Y}|\tau, \boldsymbol{q})p(\tau, \boldsymbol{q})}{p(\boldsymbol{Y})} \propto p(\boldsymbol{Y}|\tau, \boldsymbol{q})p(\tau, \boldsymbol{q}). \tag{12}$$

There are several common phylogenetic analysis software for Bayesian phylogenetic inference such as MrBayes (Ronquist et al., 2012), BEAST (Drummond & Rambaut, 2007), etc., which implement Markov chain Monte Carlo (MCMC) runs to samples from the phylogenetic posterior.

## B  ARTREE

ARTree (Xie & Zhang, 2023) generates a tree topology in an autoregressive way. The sequential generating process in ARTree facilitates a probabilistic model over tree topologies which archives leading results in phylogenetic inference. We introduce the tree topology generating process of ARTree as below and most of the statements are adapted from Xie & Zhang (2023).

Let $\tau_n = (V_n, E_n)$ be a tree topology with $n$ leaf nodes, where $V_n$ and $E_n$ represent the sets of nodes and edges, respectively. A predetermined order, also known as the taxa order, is assumed for the leaf nodes $\mathcal{X} = \{x_1, \ldots, x_N\}$. We begin by providing the definition of ordinal tree topologies.

**Definition 1** (Ordinal Tree Topology). *Let $\mathcal{X} = \{x_1, \ldots, x_N\}$ be a set of $N(N \geq 3)$ leaf nodes. Let $\tau_n = (V_n, E_n)$ be a tree topology with $n$ ($n \leq N$) leaf nodes in $\mathcal{X}$. We say $\tau_n$ is an ordinal tree topology of rank $n$, if its leaf nodes are the first $n$ elements of $\mathcal{X}$, i.e., $V_n \cap \mathcal{X} = \{x_1, \ldots, x_n\}$.*

The tree topology generation process begins with $\tau_3$ (the unique ordinal tree topology of rank 3). At the $n$-th step, assume we have an ordinal tree topology $\tau_n = (V_n, E_n)$ of rank $n$. The following steps are conducted to add the leaf node $x_{n+1}$ into $\tau_n$: i) Select an edge $e_n = (u, v) \in E_n$, which is then removed from $E_n$. ii) Add a new node $w$ and two additional edges, $(u, w)$ and $(w, v)$, to the tree topology $\tau_n$. iii)Insert the new leaf node $x_{n+1}$ and add an additional edge $(w, x_{n+1})$ to the tree topology $\tau_n$. Moreover, Xie & Zhang (2023) provides theoretical guarantees on the bijectiveness between the tree topology and the decision sequence $D = (e_3, \ldots, e_{N-1})$. Thanks

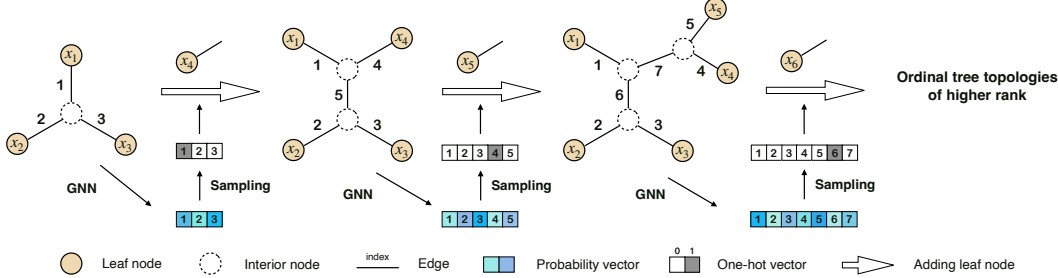

Figure 6: An illustration of building tree topology autoregressive using ARTree. This figure is from Xie & Zhang (2023).

---

**Algorithm 2:** ARTree for tree topology modeling (Xie & Zhang, 2023)

**Input:** A set $\mathcal{X} = \{x_1, \ldots, x_N\}$ of leaf nodes.
**Output:** An ordinal tree topology $\tau$ of rank $N$; the ARTree probability $Q(\tau)$ of $\tau$.
$\tau_3 = (V_3, E_3) \leftarrow$ the unique ordinal tree topology of rank 3;
**for** $n = 3, \ldots, N - 1$ **do**
    Let $f_n(u) = c_u f_n(\pi_u) + d_u$ where $\pi_u$ is the parent of $u$;
    Calculate the probability vector $q_n \in \mathbb{R}^{|E_n|}$ using the current GNN model;
    Sample an edge decision $e_n$ from Discrete $(q_n)$ and assume $e_n = (u, v)$;
    Create a new node $w$;
    $E_{n+1} \leftarrow (E_n \backslash \{e_n\}) \cup \{(u, w), (w, v), (w, x_{n+1})\}$;
    $V_{n+1} \leftarrow V_n \cup \{w, x_{n+1}\}$;
    $\tau_{n+1} \leftarrow (V_{n+1}, E_{n+1})$;
**end**
$\tau \leftarrow \tau_N$;
$Q(\tau) \leftarrow q_3(e_3) q_4(e_4) \cdots q_{N-1}(e_{N-1})$.

---

for this bijectiveness, the distribution $Q(\tau)$ over tree topologies is modelled by $Q(D)$ over decision sequences, i.e.

$$Q(\tau) = Q(D) = \prod_{n=3}^{N-1} Q(e_n | e_{<n}), \tag{13}$$

where $e_{<n} = (e_3, \ldots, e_{n-1})$ and $e_{<3} = \emptyset$. We provide the illustration and algorithm for the generating process in Figure 6 and Algorithm 2.

## C  DETAILS OF THE ENCODING MECHANISM

### C.1  PROOF OF THEOREM 1

**Theorem 1.** *Given a tree topology $\tau$ with $N$ leaf nodes, the time complexity of computing its encoding vector $s(\tau)$ is $O(N)$.*

**Proof of Theorem 1**  Assume the tree topology $\tau = (V, E)$ is stored in the binary tree data structure, where each node other than the root node also has a parent node pointer (the root node is arbitrarily selected from the internal nodes for the storage aim). Before computing the encoding of $\tau$, we first perform a postorder traversal over $\tau$ and construct an index dictionary $D$ of the mappings $\{(n, x_n)\}_{n=1}^N \cup \{(k, v)\}_{N < k < 2N-1, v \in V \backslash \mathcal{X}}$, whose time complexity is $O(N)$.

In the $n$-indexed step of the decomposition loop, we want to remove the leaf node $x_n$, which can be located by indexing the dictionary $D$ in $O(1)$ time. Following the $x_n$'s parent node pointer, we reach $x_n$'s parent $t_n$. We then detach $x_n, t_n$ from the tree and connect $t_n$'s remaining two neighbors $u_n, v_n$ whose indexes are $k_n^1, k_n^2$. Since all these local modifications of tree topologies can be done in $O(1)$ time, the time complexity of the decomposition loop is $O(N)$.

---

**Algorithm 3:** Converting encoding vectors to tree topologies

---

**Input:** A sequence $s = (s_3, \ldots, s_{N-1}) \in \mathbb{N}^{N-3}$.
**Output:** A tree topology $\tau$ with $N$ leaf nodes.
$\tau_3 \leftarrow$ the unique tree topology with the first three leaf nodes $\{x_1, x_2, x_3\}$;
$\text{Index}(x_1) \leftarrow 1; \text{Index}(x_2) \leftarrow 2; \text{Index}(x_3) \leftarrow 3; r \leftarrow$ the unique internal node of $\tau_3$;
**for** $n = 3, \ldots, N-1$ **do**
    $v_n \leftarrow$ the node in $\tau_n$ that satisfies $\text{Index}(v_n) = s_n$;
    $u_n \leftarrow$ the neighbor node of $v_n$ (towards the direction of $r$) in $\tau_n$;
    Create an node $t_{n+1}$;
    Attach the pendant edge $(t_{n+1}, x_{n+1})$ to $(u_n, v_n)$;
    $\text{Index}(t_{n+1}) \leftarrow N + n - 1; \text{Index}(x_{n+1}) \leftarrow n + 1$;
**end**

---

In the $n$-indexed step of the reconstruction loop, we have to locate the edge $(u_n, v_n)$ to add the node $x_n$. This can be done in $O(1)$ time by indexing $k_n^1, k_n^2$ in $D$. Afterwards, we de-connect $u_n, v_n$ and add $t_n, x_n$ to the graph. Since all these local modifications of tree topologies can be done in $O(1)$ time, the time complexity of the reconstruction loop is $O(N)$. Please note that the indexes in dictionary $D$ are different from the $\text{Index}(\cdot)$ function.

Therefore, the time complexity of computing the encoding vector $s(\tau)$ is $O(N)$.

### C.2 FROM ENCODINGS TO TREE TOPOLOGIES

The encoding mechanism also provides a mapping from encodings to tree topologies, which is summarized in Algorithm 3. Algorithm 3 is similar to the reconstruction loop in Algorithm 1, and has time complexity $O(N)$.

## D EXPERIMENTAL DETAILS

For all experiments, PhyloVAE is implemented in PyTorch (Paszke et al., 2019). The optimizer is Adam (Kingma & Ba, 2015) with parameters $(\beta_1, \beta_2) = (0.9, 0.999)$ and `weight_decay = 0.0`. The results are collected after 200000 iterations with batch size $B = 10$.

### D.1 FIVE-LEAF TREE TOPOLOGIES

The dimension of the latent space is set to $d = 2$. The generative model is a three-layer MLP with 512 hidden units and a ResNet architecture. For the inference model, the number of message passing rounds is $L = 2$, and both $\text{MLP}_\mu$ and $\text{MLP}_\sigma$ are composed of a two-layer MLP with 100 hidden units. The number of particles in the multi-sample lower bound (3) is $K = 32$. The learning rate is set to 0.0003 at the beginning and anneals according to a cosine schedule. The experiments are run on a single 2.4 GHz CPU.

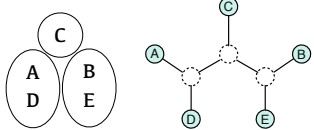

Figure 7: An illustration of the partition presentation (left) of a tree topology (right) containing five leaves {A,B,C,D,E}.

### D.2 EIGHT-LEAF TREE TOPOLOGIES

The dimension of the latent space is set to $d = 2$. The generative model is a three-layer MLP with 512 hidden units and a ResNet architecture. For the inference model, the number of message passing rounds is $L = 2$, and both $\text{MLP}_\mu$ and $\text{MLP}_\sigma$ are composed of a two-layer MLP with 100 hidden units. The number of particles in the multi-sample lower bound (3) is $K = 32$. The learning rate is set to 0.0003 at the beginning and anneals according to a cosine schedule. The experiments are run on a single 2.4 GHz CPU.

The three pre-selected tree topologies for constructing the three peaked distributions are plotted in Figure 8. Here are definitions of the Robinson-Foulds (RF) distance and the path difference (PD) distance between two tree topologies (the edge lengths are not considered).

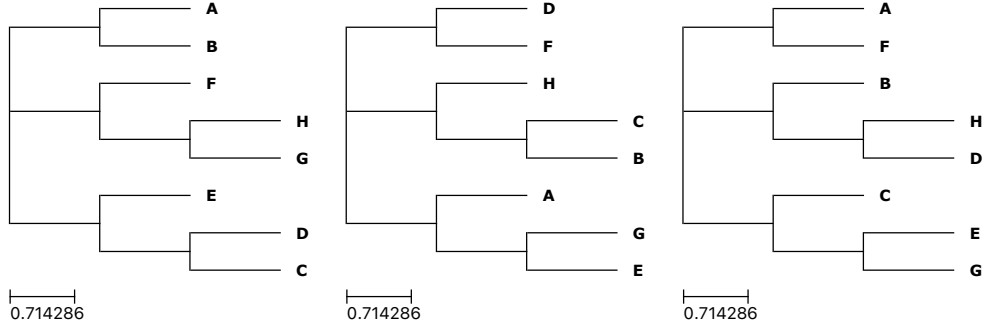

(a) The pre-selected tree topology for the first peak.

(b) The pre-selected tree topology for the second peak.

(c) The pre-selected tree topology for the third peak.

Figure 8: Pre-selected tree topologies for constructing the three-peak distribution on the 10395 tree topologies with 8 leave nodes.

Table 2: The evolutionary model for each gene in the early mammal evolution analysis (Murphy et al., 2001; Hillis et al., 2005).

| Gene | Preferred model | Base frequencies | | | | Relative substitution rates | | | | | | Pinv | Alpha |
|------|----------------|---|---|---|---|----|----|----|----|----|----|------|-------|
| | | A | C | G | T | AC | AG | AT | CG | CT | CG | | |
| ADORA3 | K2P | 0.25 | 0.25 | 0.25 | 0.25 | 1 | 3 | 1 | 1 | 3 | 1 | 0 | - |
| APP | GTR + I + $\Gamma$ | 0.25 | 0.24 | 0.18 | 0.33 | 1.6 | 3.66 | 0.47 | 0.72 | 2.65 | 1 | 0 | 0.78 |
| ZFX | HKY + I + $\Gamma$ | 0.35 | 0.23 | 0.18 | 0.23 | 1 | 7.94 | 1 | 1 | 7.94 | 1 | 0.49 | 1.24 |
| IRBP | GTR + I + $\Gamma$ | 0.21 | 0.3 | 0.3 | 0.18 | 1.5 | 4.91 | 1.34 | 0.83 | 5.8 | 1 | 0.18 | 1.04 |
| mtRNA | GTR + I + $\Gamma$ | 0.34 | 0.2 | 0.21 | 0.25 | 5.86 | 14 | 3.85 | 0.58 | 29.3 | 1 | 0.41 | 0.53 |

**Definition 2** (Robinson-Foulds distance; Robinson & Foulds (1981)). *For a tree topology $\tau = (V, E)$ with leaf nodes $\mathcal{X}$, an edge $e \in E$ divides the leaf nodes into two parts $(\mathcal{X}_1, \mathcal{X}_2)$, according to the closeness between a leaf node and the right/left terminal node of $e$. We define the unordered partition as $P_e := (\mathcal{X}_1, \mathcal{X}_2)$ and all the partitions on $\tau$ as $\mathcal{P}_\tau = \{P_e : e \in E\}$. The the Robinson-Foulds distance between two tree topologies $\tau_1$ and $\tau_2$ is defined as*

$$D_{Robinson\text{-}Foulds}(\tau_1, \tau_2) := |\mathcal{P}_{\tau_1} \Delta \mathcal{P}_{\tau_2}| = |\mathcal{P}_{\tau_1} \backslash \mathcal{P}_{\tau_2}| + |\mathcal{P}_{\tau_2} \backslash \mathcal{P}_{\tau_1}|.$$

**Definition 3** (Path difference distance; Steel & Penny (1993)). *Let $\tau$ be a tree topology with $N$ leaf nodes $\mathcal{X}$. The path length $l_{x_i, x_j}(\tau)$ between two leaf nodes $x_i, x_j$ is defined as the minimum number of edges on $\tau$ for connecting $x_i$ and $x_j$. The the path length difference distance between $\tau_1$ and $\tau_2$ is*

$$D_{path\text{-}difference}(\tau_1, \tau_2) := \sum_{1 \le i < j \le N} |l_{x_i, x_j}(\tau_1) - l_{x_i, x_j}(\tau_2)|.$$

### D.3 GENE TREES AND SEQUENCE LENGTHS

The dimension of the latent space is set to $d = 2$. The generative model is a three-layer MLP with 512 hidden units and a ResNet architecture. For the inference model, the number of message passing rounds is $L = 2$, and both $\text{MLP}_\mu$ and $\text{MLP}_\sigma$ are composed of a two-layer MLP with 100 hidden units. The number of particles in the multi-sample lower bound (3) is $K = 32$. The learning rate is set to 0.0003 at the beginning and anneals according to a cosine schedule. The experiments are run on a single 2.3 GHz CPU.

The evolutionary models of the five genes of the early mammal evolutionary analysis are in Table 2. For each gene, we simulate the DNA sequences along the ground truth phylogenetic tree in Figure 9. We also give the Newick representation of this ground truth tree (from Hillis et al. (2005)) for reproducing the results:

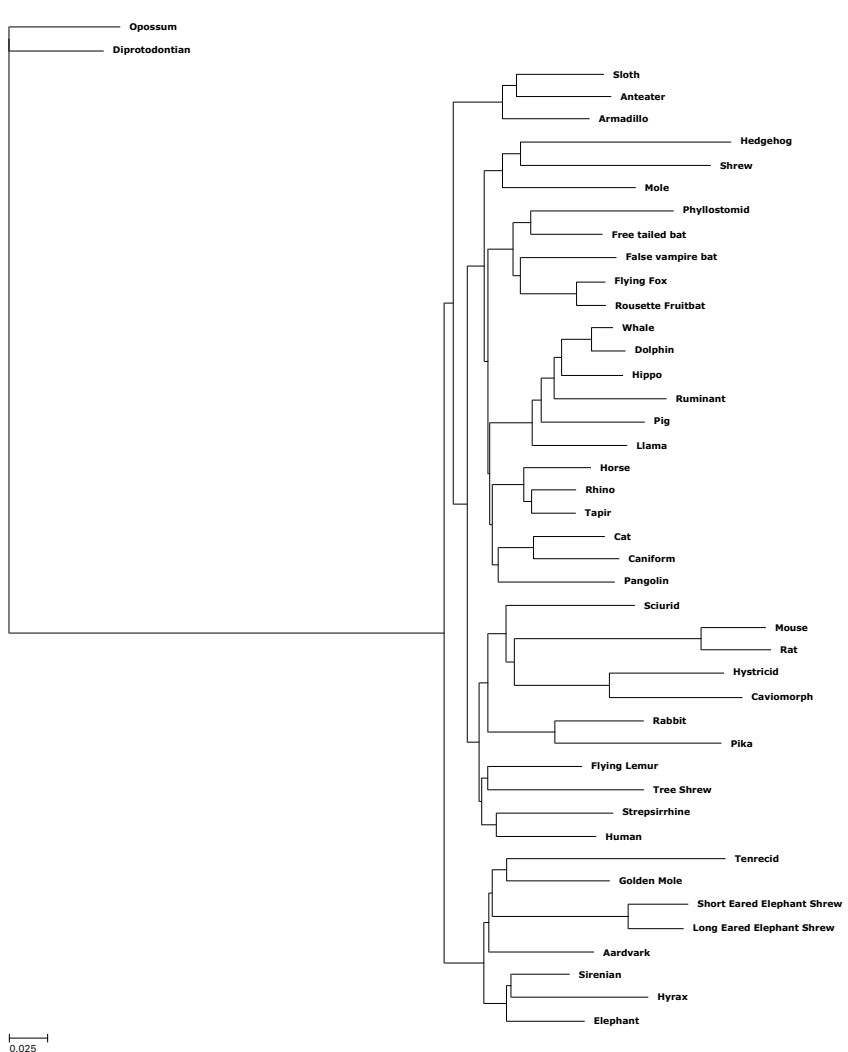

Figure 9: The ground truth phylogenetic tree for early mammal evolution (Murphy et al., 2001).

```
((Opossum:   0.072454, Diprotodontian:   0.061694):0,
((((Sloth:   0.056950, Anteater:   0.061637):0.009169,
Armadillo:  0.056660):0.032179, (((Hedgehog:   0.137379,
Shrew:  0.124147):0.011789, Mole:  0.086828):  0.011954,
(((Phyllostomid:  0.093178, Free tailed bat:  0.046665):0.011564,
(False vampire bat:  0.062583, (Flying Fox:  0.018553,
Rousette Fruitbat:  0.018931):0.036729):0.004788):0.016400,
((((((Whale:  0.013788, Dolphin:  0.021978):0.019568, Hippo:
0.039894):0.004885, Ruminant:  0.073210):  0.008450, Pig:
0.067448):0.005893, Llama:  0.061851):0.027757, ((Horse:
0.043682, (Rhino:  0.028867, Tapir:  0.028638):0.005116):0.020583,
((Cat:  0.046372, Caniform:  0.055840):0.023068, Pangolin:
0.075956):0.003871):  0.001685):0.001155):0.002432):0.011058,
(((Sciurid:  0.083962, ((Mouse:  0.042059, Rat:
0.045451):0.122018, (Hystricid:  0.074622, Caviomorph:
0.086677):0.062121):0.005432):0.011864, (Rabbit:  0.057873, Pika:
0.108683):0.043771):0.005743, ((Flying Lemur:  0.061380, Tree
Shrew:  0.101818):0.003958, (Strepsirrhine:  0.076186, Human:
0.065099):  0.009553):0.001707):0.007711):0.009175):0.005977,
```

```
((((Tenrecid:  0.142758, Golden Mole:  0.067180):0.009411, (Short
Eared Elephant Shrew:  0.039055, Long Eared Elephant Shrew:
0.036033):0.088816):  0.002240, Aardvark:  0.068518):0.003248,
((Sirenian:  0.038154, Hyrax:  0.089482):0.002916, Elephant:
0.050883):0.014801):0.025967):0.284326)
```

### D.4 MULTIPLE PHYLOGENETIC ANALYSES COMPARISON

The dimension of the latent space is set to $d = 2$. The generative model is a five-layer MLP with 512 hidden units and a ResNet architecture. For the inference model, the number of message passing rounds is $L = 2$, and both $\text{MLP}_\mu$ and $\text{MLP}_\sigma$ are composed of a two-layer MLP with 512 hidden units. The number of particles in the multi-sample lower bound (3) is $K = 32$. The learning rate is set to 0.00001 at the beginning and anneals according to a cosine schedule. The experiments are run on a single 2.3 GHz CPU.

### D.5 GENERATIVE MODELING CAPACITY ON BENCHMARK DATA SETS

Following Zhang & Matsen IV (2018); Xie & Zhang (2023), we construct the training data set and the ground truth as follows: (i) for each sequence set, there are 10 replicate training sets of tree topologies which are gathered from 10 independent MrBayes runs until the runs have ASDSF (the standard convergence criteria used in MrBayes) less than 0.01 or a maximum of 100 million iterations (tree topologies are sampled every 100 iterations with the first 25% iterations discarded); (ii) for each sequence set, the ground truth of tree topologies is gathered from 10 single-chain MrBayes for one billion iterations (tree topologies are sampled every 1000 iterations with the first 25% iterations discarded). The numbers of unique tree topologies in the training set and the ground truth for each sequence set are reported in Table 1.

We use the same model architecture and training strategy on all DS1-8. The generative model is a five-layer MLP with 512 hidden units and a ResNet architecture. For the inference model, the number of message passing rounds is $L = 2$, and both $\text{MLP}_\mu$ and $\text{MLP}_\sigma$ are composed of a two-layer MLP with 100 hidden units. The number of particles in the multi-sample lower bound (3) is $K = 32$ in Table 1 and Figure 5. The learning rate is set to 0.0003 at the beginning and anneals according to a cosine schedule. The experiments are run on a single NVIDIA RTX 2080Ti GPU.

## E ADDITIONAL EXPERIMENTAL RESULTS

### E.1 GENE TREES AND SEQUENCE LENGTHS

Figure 10 depicts the latent representations of the mammal gene trees inferred from DNA sequences with a length of 5000. We see that compared to shorter DNA sequences (Figure 4), phylogenetic inference with longer DNA sequences better discovers the ground truth tree. Again, the ADORA3 gene performs best for phylogenetic discovery, as the gene trees collapse to the ground truth tree. The MDS plots in the same setting are shown in Figure 11, where we see again that the tree topologies within a group tend to collapse towards the origins and are not sufficiently distinguished.

### E.2 MULTIPLE PHYLOGENETIC ANALYSES COMPARISON

The right plot of Figure 4 provides an example that multiple phylogenetic analyses do not converge. Here we provide an example that multiple phylogenetic analyses quickly converge.

We consider the fossil data sets (Wright, 2017) which contain 42 sequences (i.e., 42 leaf nodes) with a length of 87, which describes the biological characters of 42 types of Paleozoic crinoids. We assume the JC evolutionary model, implement two independent MrBayes (Ronquist et al., 2012) runs for 10 million iterations, and gather the tree samples per 1000 iterations from the last 3 million iterations. These 6000 tree topologies with uniform weights constitute the training set of PhyloVAE.

We find that 6000 tree topologies in the training set are distinct, which reflects the diffuse posterior of fossil phylogenies. Despite this, the latent representations of tree topologies from the two independent

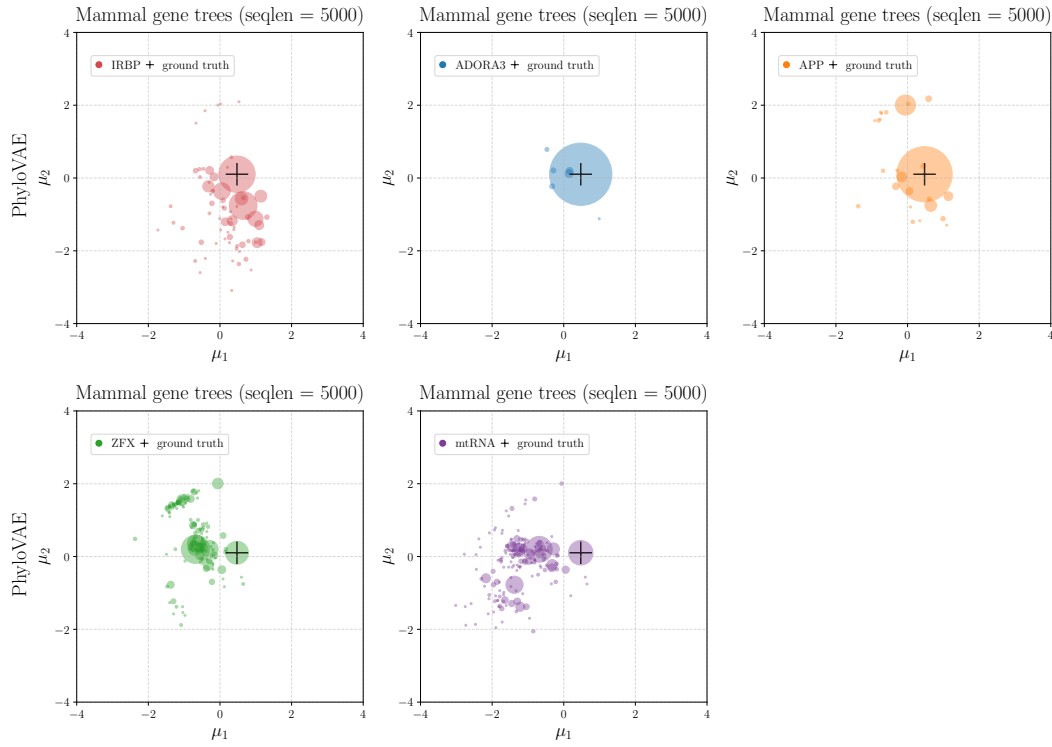

Figure 10: Latent representations obtained by PhyloVAE for the mammal gene trees inferred from DNA sequences with a length of 5000.

| KL divergence | $K = 1$ | $K = 16$ | $K = 32$ | $K = 64$ |
|---|---|---|---|---|
| $d = 2$ | 0.1275 | 0.0308 | 0.0273 | 0.0264 |
| $d = 5$ | 0.0951 | 0.0182 | 0.0177 | 0.0166 |
| $d = 10$ | 0.0997 | 0.0230 | 0.0189 | 0.0175 |

(a) DS1

| KL divergence | $K = 1$ | $K = 16$ | $K = 32$ | $K = 64$ |
|---|---|---|---|---|
| $d = 2$ | 0.0202 | 0.0097 | 0.0100 | 0.0097 |
| $d = 5$ | 0.0202 | 0.0103 | 0.0099 | 0.0107 |
| $d = 10$ | 0.0202 | 0.0107 | 0.0098 | 0.0103 |

(b) DS2

| KL divergence | $K = 1$ | $K = 16$ | $K = 32$ | $K = 64$ |
|---|---|---|---|---|
| $d = 2$ | 0.0674 | 0.0482 | 0.0529 | 0.0559 |
| $d = 5$ | 0.1397 | 0.0461 | 0.0502 | 0.0532 |
| $d = 10$ | 0.0980 | 0.0453 | 0.0477 | 0.0515 |

(c) DS3

| KL divergence | $K = 1$ | $K = 16$ | $K = 32$ | $K = 64$ |
|---|---|---|---|---|
| $d = 2$ | 0.1038 | 0.0646 | 0.0619 | 0.0607 |
| $d = 5$ | 0.0995 | 0.0470 | 0.0467 | 0.0471 |
| $d = 10$ | 0.1082 | 0.0470 | 0.0469 | 0.0460 |

(d) DS4

Table 3: KL divergence of PhyloVAE to the ground truth on DS1-4 with varing $d$ and $K$.

MrBayes runs seem to have the same distribution. This implies the underlying convergence of the two MrBayes runs.

### E.3 GENERATIVE MODELING ON BENCHMARK DATA SETS

Table 3 reports the KL divergence obtained by different choices of $d$ and $K$ on DS1-4. We see that increasing $K$ can generally improve the approximation accuracy, while sometimes a large $d$ may increase the training difficulty and lead to overfitting.

Figure 13 compares the probability estimates and the ground truth. We see that PhyloVAE provides reliable probability estimation.

Figure 14 shows latent representations of the tree topologies in the training set (repository 1) of DS1-8 produced by PhyloVAE.

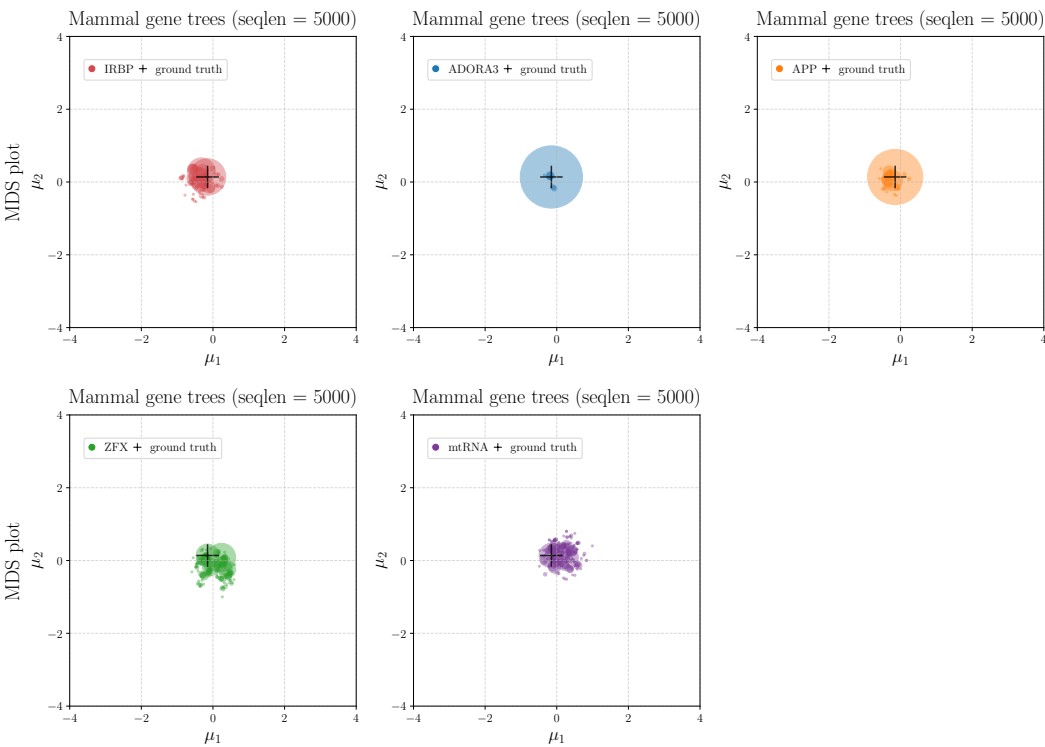

Figure 11: MDS plots for the mammal gene trees inferred from DNA sequences with a length of 5000.

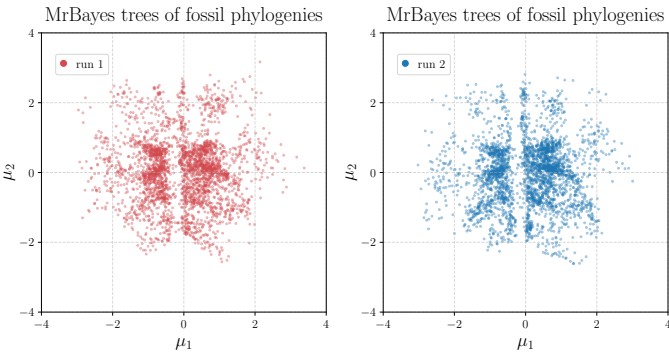

Figure 12: Latent representation of two groups of tree topologies gathered from independent MrBayes runs on the fossil data sets.

To more comprehensively evaluate the efficiency of PhyloVAE, we report the memory usage of it and the baseline method ARTree in Table 4.

Table 4: Memory usage (MB) of running ARTree and PhyloVAE ($d = 10, K = 32$) on DS1-8.

| Data set | DS1 | DS2 | DS3 | DS4 | DS5 | DS6 | DS7 | DS8 |
|---|---|---|---|---|---|---|---|---|
| ARTree | 400.37 | 428.79 | 518.84 | 600.50 | 784.32 | 778.05 | 932.56 | 1067.93 |
| PhyloVAE ($d = 10, K = 32$) | 635.67 | 637.72 | 655.69 | 785.75 | 689.15 | 701.30 | 784.41 | 818.72 |

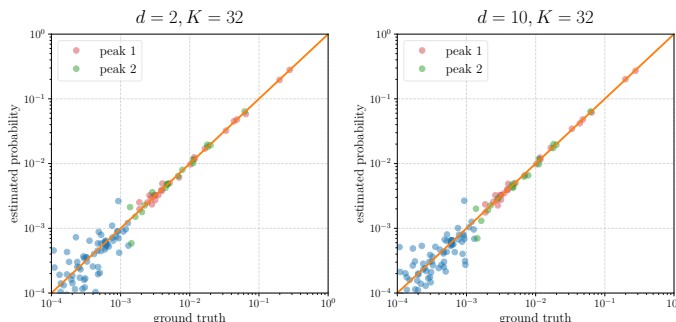

Figure 13: Tree probability estimates obtained by PhyloVAE v.s. ground truth on DS1.

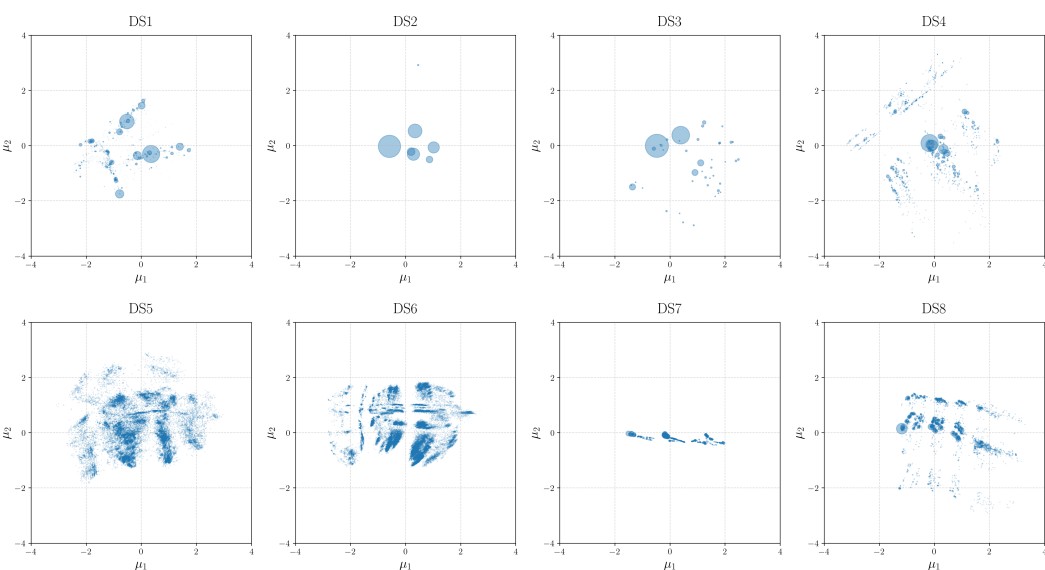

Figure 14: Latent representation of the tree topologies in the training set (repository 1) of DS1-8. The scatter size is proportional to the probability of the tree topology.

