# OpenReview forum: "PhyloVAE: Unsupervised Learning of Phylogenetic Trees via Variational Autoencoders"
_ICLR.cc/2025/Conference — ICLR 2025 Poster_

### Official Review · Reviewer_7Ksu · 2024-10-31

**Soundness:** 4
**Presentation:** 4
**Contribution:** 3
**Rating:** 8
**Confidence:** 4

**Summary:**

The authors propose a VAE for learning latent representations of phylogenetic tree topologies and generative processes thereof. Arguably the main contribution lies in the clever encoding structure that runs in the order of leaf nodes. A crux to making the VAE work for trees is to use pre-computed tree topologies as training data (inputs to the encoder). They experiment on standard and less standard real phylogenetic data.

**Strengths:**

This is a very nice submission that is well-written and which proposes a clever approach to learn representation and generative processes of phylogenetic trees. The experimental results are convincing and significant, especially taking into account the run times compared to ARTree. It utilizes previous results and ideas from the VI-based phylogenetic tree inference literature to, in a non-trivial way, devise a VAE algorithm.

**Weaknesses:**

* It is appreciated that it is clearly stated that the inputs to the encoder are collections of topologies. In [1; Fig. 5] the posterior coverage of the pre-computed topologies was evaluated and compared with those obtained from MrBayes. Can you provide a similar quantification? It might even suffice to merely reference their work to guard aganst the common suspicion that pre-computed tree topoplogies are not good representations.

* On the same topic, in Sec. 5.2 the topologies are gathered from a 1 milion MCMC run, but this is not this quite short, in order to ensure that the samples are coming from the posterior?  To be clear, I do not think that it is an issue to learn posteriors over a pre-computed set of topologies, if the set of topologies is a good representation of the true posterior's support.

* The collection of trees are tuples of the topologies and their corresponding weights. The weights are then in a footnote described to be the frequency of the topology (right?). Then in all experiments, the weights are explicitly mentioned to be uniform. Why is this? Is a topology never sampled more than once in the pre-processing stage? What would happen if the trees were weighted by scores obtained, for instance, from MrBayes? Or simply the Felsenstein likelihood of the topology, normalized over all samples.

* Recently there have been works for applying mixtures of variational distributions in VAEs [2], phylogenetic posterior inference [3] as well as in VAEs *and* phylogenetics [4] (note that [4] does not propose a VAE for phylogenetics, but applies mixtures in both settings). I think at least [3] and [4] should be mentioned in the list of works referenced at line 463 to emphasize the great, current interest in Bayesian phylogenetics at ML conferences.

## Minor
* There is a clash of notations for $w$ notating both internal node connections in Sec. 3.2 and the weights of the tree topologies.

* I would be good to disambiguate the proposed method to other tree-based VAEs such as [5, 6] and [7] (although [7] appears to not be published). None of these references experiment on phylogenetic data, but the originality of the proposed method would be emphasized by contrasting your encoder and generative processes to these works.

[1] https://jmlr.org/papers/volume25/22-0348/22-0348.pdf

[2] https://proceedings.mlr.press/v202/kviman23a.html

[3] https://arxiv.org/abs/2310.00941

[4] https://arxiv.org/abs/2406.07083

[5] https://arxiv.org/pdf/2306.08984

[6] https://arxiv.org/pdf/1810.06891

[7] https://openreview.net/forum?id=Hy0L4t5el

**Questions:**

* Relating to the references provided regarding mixtures of variational distributions, I am wondering if your proposed method could benefit from the mixture techniques provided in [2,3,4]? Potentially they could help PhyloVAE to outperform ARTree in Table 1? In [4] the estimators make the learning of the mixture components efficient in terms of run time.

* Is Algorithm 1 parallellizable on a GPU? Can you use batches of tree topologies during training?

---

> ### Author Response · Authors · 2024-11-22
> **Response to Reviewer 7Ksu's concerns**
>
> Thanks for your constructive feedback! Here are our responses.
>
> **W1**: It is appreciated that it is clearly stated that the inputs to the encoder are collections of topologies. (...)
>
> **Response to W1**: Thanks for your suggestion! First, we'd like to clarify that the collections of tree topologies as inputs to PhyloVAE are for representation learning tasks, where these collections are just the data sets whose (low-dimensional) representations are of interest (e.g., for visualization). In [1] and other variational approaches for Bayesian phylogenetics, a collection of pre-computed tree topologies is also required. However, it is used for subsplit support estimation in [1], which is essential for the parameterization of SBNs. In fact, PhyloVAE can also be used for variational Bayesian phylogenetic inference. Similarly to ARTree, PhyloVAE naturally specifies a family of distributions over the entire tree topology space, without requiring pre-computed tree topologies for support estimation.
> Therefore, we do not need to investigate the efficiency of support estimation as in [1: Figure 5].
> One advantage of these support-free VI methods for Bayesian phylogenetics is that they can be applied when support estimation becomes challenging (e.g., diffuse posteriors as discussed in [1] as well).
> When support estimation can be done efficiently (e.g., good collections of pre-computed tree topologies are available/easy to obtain, using methods suggested in [1]), previous methods that require support estimation can also work well.
> We will clarify this in our revision.
>
>
> **W2**: On the same topic, in Sec. 5.2 the topologies are gathered from a 1 milion MCMC run, (...)
>
> **Response to W2**: The choice of 1 million simply follows the setting in [Hillis et al., 2005] who considered the same experiment.
> We have clarified this in our revision.
>
>
> **W3**: The collection of trees are tuples of the topologies and their corresponding weights. The weights are then in a footnote described to be the frequency of the topology (right?). Then in all experiments, the weights are explicitly mentioned to be uniform. Why is this? Is a topology never sampled more than once in the pre-processing stage? What would happen if the trees were weighted by scores obtained, for instance, from MrBayes? Or simply the Felsenstein likelihood of the topology, normalized over all samples.
>
> **Response to W3**: Trees from MrBayes runs can be sampled more than once. They (potentially duplicated) are assigned with uniform weight first, and then the duplicated trees are merged and re-assigned with weights proportional to the number of duplicated trees.
> For ease of description, we do not describe the merging operation explicitly.
> In a word, the actual weights of unique trees are indeed the empirical frequencies.
>
>
> **W4**: Recently there have been works for applying mixtures of variational distributions (...)
>
> **Response to W4**: Thank you for providing these related works. We will reference these works in our revision.
>
>
> **W5**: There is a clash of notations $w$ for notating both internal node connections in Sec. 3.2 and the weights of the tree topologies.
>
> **Response to W5**: Thanks for the nice catch! We will use different notations.
>
> **W6**: I would be good to disambiguate the proposed method to other tree-based VAEs (...)
>
> **Response to W6**:
> Although [5,6,7] contain "tree" and "VAE" in their titles, they consider a tree-shaped prior distribution or hierarchical latent variable structure. These papers do not consider modeling any graph or tree objects and thus are clearly distinct from our PhyloVAE.
> We have clarified this in our revision.
>
> **Q1**: Relating to the references provided regarding mixtures of variational distributions, I am wondering if your proposed method could benefit from the mixture techniques provided in [2,3,4]?
>
> **Response to Q1**: For variational inference, the mixture models indeed provide a more powerful variational distribution. Therefore, we can expect improvement in approximation accuracy if applying mixture techniques to PhyloVAE. Note that PhyloVAE itself can be viewed as a mixture model with infinitely many components.
>
> **Q2**: Is Algorithm 1 parallellizable on a GPU? Can you use batches of tree topologies during training?
>
> **Response to Q2**: The answer is yes. Algorithm 1 is parallelizable on a GPU.

---

> > ### Comment · Reviewer_7Ksu · 2024-11-22
> >
> > Thank you for your response. Most if not all of my questions were answered, although I have some follow-up questions.
> >
> > **W1**: Interesting, and thanks for the explanation. I look forward to seeing a brief discussion on how to choose candidate-tree proposals in your setting (I couldn't find this in the revision?)
> >
> > **W2**: Well, although the number of MCMC iterations are the same as in an experiment in the reference, it doesn't really answer my question about how sufficient this number is for ensuring that samples are drawn proportionally to the posterior. I am afraid that the text in this submission might be parsed as that 1 million iterations being sufficient, while no conviction of this is provided. To be clear, a verbal reflection may be enough.
> >
> > **W3**: I see, thanks for the explanation. While I agree that there is no need for a long explanation of the weighting scheme, I sincerely think that the explanation in the footnote can be rephrased to be more clear by saying that the weight of a tree topology is the frequency among the MrBayes samples.
> >
> > **Q1**: This is interesting. Could you explain how PhyloVAE can be viewed as an infinite mixture?
> >
> > Thanks again for your response!

---

> ### Author Response · Authors · 2024-11-24
> **Thanks for your follow-up questions!**
>
> Thanks for your follow-up questions! Here are our responses:
>
> **Response to W1**
> Thanks for the follow-up question.
> PhyloVAE mainly aims to learn representations of a pre-given data set of tree topologies, no matter how they are collected.
> These data sets often come from phylogenetic analysis softwares such as MrBayes and BEAST, but they can also come from observations, open-source data sets, and other biological softwares, as long as they are of scientific interest.
> What we want to emphasize is that the source of data, the parameters/proposals of the software, etc, are orthogonal of PhyloVAE's task and will not affect the effectiveness of PhyloVAE.
> (Even if a poor proposal is configured, PhyloVAE is still able to diagnose the underlying divergence of MCMC run, see section 5.2)
> We added these discussions to footnote 2 on page 3 in our revision.
>
> **Response to W2**
> To assess the convergence of MrBayes runs, we report the ASDSF at the 1,000,000-th iteration. We see all of these runs have an ASDSF less than 0.01, indicating the underlying convergence. We have clarified this in Section 5.2.
>
> *Table: ASDSF of MrBayes run at the 1,000,000-th iteration (seqlen=1000)*
> |Gene|ADORA3|APP|IRBP|mtRNA|ZFX|
> |---|---|---|---|---|---|
> |ASDSF|0.0029|0.0028|0.0033|0.0026|0.0039|
>
> **Response to W3**
> Thanks for this suggestion. We have modified the line 157 accordingly.
>
> **Response to Q1**
> The marginal probability of tree topology $\tau$ given by PhyloVAE takes the form $p_\theta(\tau)=\int_{\mathbb{R}^d}p_\theta(\tau|z)p(z)dz = \lim_{n\rightarrow\infty}\frac1n \sum_{i=1}^n p_\theta(\tau|z_i),\; z_i\sim p(z)$. Here the prior $p(z)$ can be viewed as a continuous mixing distribution, leading to a mixture model with infinitely many components.
>
> Please feel free to let us know if you have any further questions!

---

### Official Review · Reviewer_JV6s · 2024-11-02

**Soundness:** 3
**Presentation:** 3
**Contribution:** 3
**Rating:** 6
**Confidence:** 4

**Summary:**

The authors introduce a cheap approximation of the ARTree generative model for phylogenetic trees (by assuming independence of the actions executed sequentially to construct a tree). This allows faster training compared to ARTree because generation (or reconstruction variational bound) can be done in parallel (over all tree construction actions) rather than sequentially, which yields a runtime speedup on modern computers. They evaluate the new model (PhyloVAE) for the purpose of 2D visualization and quantify the loss in modeling power (in terms of KL divergences to the ground truth).

I have increased my ratings after reading the rebuttal.

**Strengths:**

Phylogenetic trees are interesting objects to model and visualize and it could be the case that training on vary large collection of very large trees could be computationally expensive with previous methods such as ARTree.

**Weaknesses:**

(1) to be convincing, the paper should show a case where ARTree is really too slow for being practical

(2) The experiments reported take on the order of seconds or less than a minute to train, so one wonder if the gain in computing power is so useful, if that would be at the expense of worse modeling of the tree distributions.

(3) Although two different capacities were tested for PhyloVAE (dimension 2 and 10), that was not the case for ARTree, making it unclear if ARTree could have obtained a better KL divergence with a different capacity.

(4) In terms of modeling power, assuming independence of the actions seems very strong. It is not clear that it will work well for other tree distributions not tested in this paper.

**Questions:**

Please try to address the weaknesses in the above section.

---

> ### Author Response · Authors · 2024-11-22
> **Response to Reviewer JV6s's concerns**
>
> Thanks for your careful review. Here are our responses to your concerns.
>
> **(1)** to be convincing, the paper should show a case where ARTree is really too slow for being practical.
>
> **Response to (1)**:
> Please see Figure 5 for computational times. We want to clarify that although it costs only several seconds for ARTree per 10 iterations, the total training process contains 200,000 iterations! The total training time for reproducing the results in Section 5.3 is reported in the following Table. We see that it costs more than 100 hours to train ARTree on DS7 and DS8.
>
> *Table: Total training time (hours) of ARTree and PhyloVAE in Section 5.3.*
>
> |Training time (hours)|DS1|DS2|DS3|DS4|DS5|DS6|DS7|DS8|
> |---|---|---|---|---|---|---|---|---|
> |ARTree|33.54|33.87|50.4|63.82|71.23|76.87|113.2|128.2|
> |PhyloVAE|3.7|2.9|3.67|4.59|5.82|5.36|8.8|9.1|
>
>
> **(2)** The experiments reported take on the order of seconds or less than a minute to train, so one wonder if the gain in computing power is so useful, if that would be at the expense of worse modeling of the tree distributions.
>
> **Response to (2)**:
> Figure 5 reports the training time of **10 iterations**, but the total training process requires **200,000 iterations**. The total training time is reported in the above table, where we see that it costs more than 100 hours to train ARTree on DS7 and DS8.
>
>
> **(3)** Although two different capacities were tested for PhyloVAE (dimension 2 and 10), that was not the case for ARTree, making it unclear if ARTree could have obtained a better KL divergence with a different capacity.
>
> **Response to (3)**:
> Firstly, we'd like to emphasize that we do not expect an accuracy gain upon baselines, as the major advantages of PhyloVAE are **representation learning ability** and **fast computation speed**.
> Moreover, we tried to use enlarge the ARTree by employing more layers ($L=4$) in message passing. Although $L=4$ achieves a better KL on DS1, it can lead to worse results on DS3 and DS4, which can be attributed to the increased training difficulty or the potential over-fitting problem.
>
> *Table KL divergence with different number of layers in the message passing step of ARTree*
> |Number of layers in message passing of ARTree \ Data set| DS1 | DS2 | DS3 | DS4 |
> | --- | --- | --- | --- | --- |
> |$L=2$|0.0045|0.0097|0.0548|0.0299|
> |$L=4$|0.0030|0.0099|0.0767|0.0354|
>
>
> **(4)** In terms of modeling power, assuming independence of the actions seems very strong. It is not clear that it will work well for other tree distributions not tested in this paper.
>
> **Response to (4)**:
> We would like to emphasize that we assume **conditional independence** instead of independence among actions.
> Combining a mixing prior distribution $p(z)$ and a conditionally independent $p(s(\tau)|z)$, PhyloVAE has enough capacity to model a complicated marginal distribution $p(s(\tau))$.
> To see this, consider a standard VAE for image data, where $p(z)$ is standard Gaussian, and $p(x|z)$ is a conditional Gaussian with **diagonal covariance matrix**, but this can model complicated image data.
>
> About the additional tree distributions, DS1-8 are the most commonly considered benchmark in phylogenetics [1,2,3,4,5].
> Moreover, it has been shown in [6] that DS1-8 have all sorts of strange aspects, including multiple modes separated by a difficult-to-cross valley, and duplicated substructures caused by lack of resolution in the data.
> Therefore, we can expect PhyloVAE to work for other distributions if it works for DS1-8.
> We will clarify this in our revision.
>
> [1] Zhang, C. and Matsen IV, F. A. "Variational Bayesian phylogenetic inference." International Conference on Learning Representations (2019).
>
> [2] Koptagel, Hazal, et al. "Vaiphy: a variational inference based algorithm for phylogeny." Advances in Neural Information Processing Systems 35 (2022): 14758-14770.
>
> [3] Mimori, Takahiro, and Michiaki Hamada. "GeoPhy: differentiable phylogenetic inference via geometric gradients of tree topologies." Advances in Neural Information Processing Systems 36 (2023).
>
> [4] Xie, T. and Zhang, C. "ARTree: A deep autoregressive model for phylogenetic inference." Advances in Neural Information Processing Systems 36 (2023).
>
> [5] Zhou, Mingyang, et al. "PhyloGFN: Phylogenetic inference with generative flow networks." arXiv preprint arXiv:2310.08774 (2023).
>
> [6] Chris Whidden and Frederick A Matsen IV. Quantifying MCMC exploration of phylogenetic tree space. Systematic Biology, 2015.

---

### Official Review · Reviewer_vhRJ · 2024-11-03

**Soundness:** 3
**Presentation:** 3
**Contribution:** 3
**Rating:** 6
**Confidence:** 4

**Summary:**

The paper introduces PhyloVAE, an unsupervised learning framework for phylogenetic tree topologies that utilizes variational autoencoders (VAE). PhyloVAE is designed to both learn informative representations of phylogenetic tree structures and generate new tree topologies efficiently. The model combines a novel encoding mechanism inspired by autoregressive topology generation, creating a deep latent-variable model that maps arbitrary points in the latent space to tree topologies, enabling high-resolution and parallelized topology generation. The framework demonstrates its effectiveness in representation learning and generative modeling, outperforming traditional distance-based methods by providing both visualization and probabilistic modeling for tree topology distributions.

**Strengths:**

1. Highly Relevant and Practical Topic: The paper addresses the generation and representation of phylogenetic trees, a critical problem in evolutionary and computational biology. By introducing an unsupervised generative model, the paper offers a novel approach for analyzing complex evolutionary relationships. This work is practically significant as it enables researchers to gain deeper insights into biological evolutionary paths and diversity, advancing the field substantially.
2. Significant Efficiency Gains: PhyloVAE achieves efficient phylogenetic tree generation through its non-autoregressive design, enabling faster and parallelized tree structure generation. Compared to traditional stepwise generation methods, this improvement is especially meaningful for handling large-scale biological data, accelerating the computational process and enhancing the model’s feasibility and applicability in real-world scenarios.

**Weaknesses:**

1. Need for Generalizability Testing Across Datasets: The paper primarily focuses on evaluating PhyloVAE across different benchmark datasets (DS1-DS8) without testing its generalizability across distinct datasets. For example, testing a model trained on DS1-DS4 and then assessing its performance on DS5 would offer insight into the model’s robustness across varied evolutionary patterns. Such cross-dataset evaluations are crucial for demonstrating the model’s applicability beyond specific training sets.
2. Lack of Comparison with True Phylogenetic Trees: While PhyloVAE performs well in capturing tree structures, there is no direct comparison between the generated phylogenetic trees and real, experimentally derived species trees. This comparison would help validate the biological relevance of the model’s output by assessing how closely PhyloVAE’s generated trees align with known evolutionary relationships. In-depth analysis of these discrepancies would offer valuable insights into the model’s accuracy.
3. Marginal Improvements in Performance Metrics: The paper demonstrates efficiency gains in tree generation time but does not show substantial improvements in accuracy metrics compared to state-of-the-art methods. As shown in Table 1, while PhyloVAE’s performance is comparable, it does not significantly outperform traditional models like ARTree in KL divergence to ground truth trees. This raises questions about the trade-offs made between efficiency and accuracy in the model’s design.

**Questions:**

1. How does PhyloVAE perform when trained on one set of datasets (e.g., DS1-DS4) and tested on a distinct dataset like DS5? Could the authors provide insights or additional experiments to demonstrate the model’s robustness across varied evolutionary patterns? Cross-dataset evaluations would enhance the assessment of PhyloVAE’s broader applicability. To make this clearer, could the authors also analyze specific aspects of the datasets—such as the number of species, evolutionary rates, or tree shapes—that might influence generalization performance? Additionally, specific metrics or visualizations for comparing performance across datasets would help clarify how different components, like the encoding mechanism or the inference model, contribute to the model’s generalizability.
2. Has PhyloVAE been directly compared to real, experimentally derived species trees? Such a comparison could provide essential validation for the model’s biological relevance by examining how closely the generated trees align with established evolutionary relationships. Could the authors explore potential discrepancies between PhyloVAE-generated trees and actual species trees and analyze their implications for the model’s accuracy? Including specific datasets or types of experimentally derived trees, as well as visualization techniques or metrics, would help quantify and illustrate these similarities and differences.
3. While PhyloVAE offers efficiency gains, it appears not to outperform traditional models like ARTree in terms of accuracy metrics, as noted in Table 1’s KL divergence comparisons. Could the authors clarify the trade-offs made between efficiency and accuracy in the model design, and whether further refinements might enhance both speed and accuracy? A detailed analysis of how specific model components or hyperparameters affect this trade-off would be helpful. For example, it would be insightful to explore how increasing the latent space dimension or adjusting the number of particles in importance sampling impacts both computational efficiency and accuracy (e.g., KL divergence). These investigations could offer valuable directions for potential model refinements.

---

> ### Author Response · Authors · 2024-11-22
> **Response to Reviewer vhRJ's concerns**
>
> Thanks for your careful review! Here are our responses to your concerns.
>
> **W1**: Need for Generalizability Testing Across Datasets
>
> **Response to W1**:
> Like general VAE models, in the current format, we do not expect PhyloVAE to provide universal representation across distinct datasets, as the models are trained specifically for each dataset.
> It's like doing a PCA on a data set, where the coefficients of principle components are also dataset-specific.
> However, we admit that exploring the generalization ability of representation models is valuable and will consider this in future works (maybe a sequence embedding scheme is needed).
>
>
> **W2**: Lack of Comparison with True Phylogenetic Trees
>
> **Response to W2**:
> We would like to clarify that all the phylogenetic trees are from statistical inference, instead of biological experiments.
> In fact, the underlying “biological” phylogenetic trees are never available except in some very unusual circumstances (e.g., directed evolution experiments).
> As a submission to an AI conference, this paper understands phylogenetics as a computational and statistical problem, instead of a biological problem.
> From this perspective, we construct the "ground truth" posterior trees by running sufficiently long-run MrBayes, which is a common practice in this field.
> Afterward, we compute the KL divergence from PhyloVAE to this ground truth, as reported in Table 1, to verify that PhyloVAE indeed generates good posterior trees.
>
> To provide an in-depth analysis between PhyloVAE and the ground truth, we added the scatter plot between probability estimates and the ground truth, in Figure 13.
> We see that PhyloVAE indeed provides good estimates for these high-quality ground truth trees.
>
>
> **W3**: Marginal Improvements in Performance Metrics
>
> **Response to W3**:
> Thanks for pointing this out. We agree that PhyloVAE shows marginal improvement upon ARTree in terms of KL divergence, however this is not the central advance reported here. However, the superiority is summarized in the following two aspects:
> - As specifically described in the Introduction, **the main focus of PhyloVAE is the representation learning**, and PhyloVAE is **the first** deep model to do this. We also conducted extensive experiments to support this point.
> - PhyloVAE shows a great improvement in terms of computational efficiency, as shown in Figure 5. This is the most critical advantage of PhyloVAE for generative modeling.
>
> **Q1**: How does PhyloVAE perform when trained on one set of datasets (e.g., DS1-DS4) and tested on a distinct dataset like DS5?
>
> **Response to Q1**: Thanks for this insightful question. We have to admit this is not possible for our current model setting. Please find a more detailed explanation in our response to W1.
>
> **Q2**: Has PhyloVAE been directly compared to real, experimentally derived species trees?
>
> **Response to Q2**:
> Please see our response to W2. In Figure 13, we provide an additional comparison between the tree probability estimates of PhyloVAE and the ground truth. This ground truth is statistically inferred from a long-run MrBayes, which is a common practice in this field.
>
> **Q3**: While PhyloVAE offers efficiency gains, it appears not to outperform traditional models like ARTree in terms of accuracy metrics (...)
>
> **Response to Q3**:
> About the minor improvement of KL, we would like to emphasize that the main advantages of PhyloVAE lie in representation learning ability and computational efficiency.
> Please see our response to W3 for a detailed explanation.
>
> To show the trade-off between effectiveness and efficiency, we provided an additional ablation study on $d$ and $K$ in Table 3 in our revision.
> We also put it here for your ease.
> We see that increasing $K$ can generally improve the approximation accuracy, while sometimes a large $d$ may increase the training difficulty and lead to overfitting.
>
> *Table: KL divergence on DS1*
> |KL | K=1 | K=16| K=32| K=64|
> |---|---|---|---|---|
> |d=2|0.1275|0.0308|0.0273|0.0264|
> |d=5|0.0951|0.0182|0.0177|0.0166|
> |d=10|0.0997|0.0230|0.0189|0.0175|
>
> *Table: KL divergence on DS2*
> |KL | K=1 | K=16| K=32| K=64|
> |---|---|---|---|---|
> |d=2|0.0202|0.0097|0.0100|0.0097|
> |d=5|0.0202|0.0103|0.0099|0.0107|
> |d=10|0.0202|0.0107|0.0098|0.0103|
>
> *Table: KL divergence on DS3*
> |KL | K=1 | K=16| K=32| K=64|
> |---|---|---|---|---|
> |d=2|0.0674|0.0482|0.0529|0.0559|
> |d=5|0.1397|0.0461|0.0502|0.0532|
> |d=10|0.0980|0.0453|0.0477|0.0515|
>
> *Table: KL divergence on DS4*
> |KL | K=1 | K=16| K=32| K=64|
> |---|---|---|---|---|
> |d=2|0.1038|0.0646|0.0619|0.0607|
> |d=5|0.0995|0.0470|0.0467|0.0471|
> |d=10|0.1082|0.0470|0.0469|0.0460|

---

> > ### Comment · Reviewer_vhRJ · 2024-11-27
> > **Thank you for your response and since my concerns were not completely addressed (W1 and W3), I decided to keep my scores.**
> >
> > Thank you for your response and since my concerns were not completely addressed (W1 and W3), I decided to keep my scores.

---

> > > ### Author Response · Authors · 2024-11-28
> > > **Thanks for your positive evaluation of our paper!**
> > >
> > > Thank you for your positive evaluation of our paper! Regarding the trade-off between efficiency and accuracy, Table 3 in our revised manuscript reports the results under different choices of latent dimension $d$ and number of particles $K$, which can help the readers better understand the trade-off.

---

### Official Review · Reviewer_SRrn · 2024-11-04

**Soundness:** 3
**Presentation:** 2
**Contribution:** 3
**Rating:** 6
**Confidence:** 5

**Summary:**

PhyloVAE adopts the architecture of variational autoencoders and is specially designed to handle phylogenetic tree topologies. It maps any point in the latent space to a specific tree topology, and this architecture makes it excel at distinguishing between different tree shapes, outperforming traditional distance-based methods. Moreover, PhyloVAE's generative property, which enables the mapping of any point in the latent space to a specific tree topology, is an important advantage over other models that lack this functionality. This property enhances the visualization and interpretability of the learned representation.

**Strengths:**

* **(S1)** The proposed PhyloVAE framework is highly innovative. To my knowledge, a variational autoencoder is applied to the representation learning of phylogenetic trees for the first time, demonstrating the potential of deep learning methods in this field.

* **(S2)** This paper is overall easy to follow to some extent (but also has some drawbacks, as mentioned weaknesses). The model architecture and training process are described in detail, especially the mapping function in potential space and visualization of tree topology.

* **(S3)** The effectiveness of PhyloVAE is fully verified through experiments, especially its performance in dealing with complex tree structures, which shows the advantages over traditional distance-based and density-based estimation methods. The experimental design is reasonable, and the used datasets are appropriately selected.

**Weaknesses:**

* **(W1)** Some important assertions are unclear or unsupported. The author mentioned that the *model has high-resolution representations of phylogenetic tree samples*. How was this proven through experiments and theatrical analysis? Meanwhile, the authors mention that *their effectiveness heavily depends on the choice of distance metric and can sometimes exhibit counterintuitive behaviors*. What specifically does this counterintuitive behavior refer to?

* **(W2)** The specific application scenarios described in the manuscript are vague. Although the importance of tree visualization is emphasized in the intro and abs sections, the main goal of the model is not clearly distinguished between representation learning and generative modeling. I suggest the authors further clarify the core goal of the proposed methods (or tackled problem) in the introduction section and clearly distinguish the relationship between representation learning and generative modeling. Meanwhile, the author might provide a specific discussion on the potential use of the model in different application scenarios to enhance the logic and practical significance of the manuscript.

* **(W3)** This paper mentions the limitations of distance-based and density-based estimation methods. Still, it lacks an in-depth analysis of the PhyloVAE model settings to clarify its advantages in representation learning and visualization. What model settings or methods can perform well in representation learning and visualization?

* **(W4)** More minor issues could be clarified:
  - **(a)** What is the impact of different phylogenetic inference software choices?
  - **(b)** Why is the formula for the data distribution in line 157? Are there any a priori constraints?
  - **(c)** The simulated dataset used in the experiments in Section 5.1 only contains tree topologies with five and eight-leaf nodes, and experiments on large-scale tree structures are lacking.
  - **(d)** The paper does not clearly explain the specific generation process of the tree topologies in the experiments in Section 5.1.
  - **(e)** The paper's formula 3 has a variable K, and so does line 346. Do they represent the same variable?

**Questions:**

* **(Q1)** One of the motivations mentioned in the text is: *The classical approach to visualize and analyze distributions of phylogenetic trees is to calculate pairwise distances between the trees and project them into a plane using multidimensional scaling (MDS) .* *However, these approaches have the shortcoming that one can not map an arbitrary point in the visualization to a tree, and therefore do not form an actual visualization of the relevant tree space.* However, as the main motivation of this paper, it does not explain in detail the limitations of this mapping and its specific impact on visualization and analysis. Why is this operation a disadvantage? And what are the shortcomings of the MDS method in representing tree space?

* **(Q2)** The encoding mechanism in Section 3.2 is clearly written, but there is a potential problem:

  - **(a)** In the Decomposition stage, when selecting edge e_n, the conditional probability $Q(e_n | e_{<n})$ is based on the selected edge set $e_{<n}$. As the depth of the tree and the number of edges increases, the calculation of the conditional probability may become complicated.
  - **(b)** In the Reconstruction stage, if an error occurs in the previous Decomposition process, the reconstructed tree may not be consistent with the original tree. How should this be corrected?

* **(Q3)** The paper claims that PhyloVAE is ‘the first representation learning framework that targets the topology of phylogenetic trees’, but existing methods such as VBPI-GNN also have representation learning capabilities. This statement lacks discussion and comparison with relevant existing work, making its claim of innovation seem less rigorous and may make readers question the contribution of the article. Similarly, why not compare with baselines?

* **(Q4)** Where is the hyper-parameter experiment, e.g., how to decide the parameters $d$, $K$, $L$, Iterations, and batch size?

---

> ### Author Response · Authors · 2024-11-22
> **Response to Reviewer SRrn's concerns - 1**
>
> Thanks for your constructive feedback! We addressed your concerns as follows.
>
> **W1**: Some important assertions are unclear (...)
>
> **Response to W1**:
> Thanks for your question. For the argument of high-resolution representations, please refer to Figure 4.
> In Figure 4, we see that PhyloVAE gives representations that are clearly separated, and subgroups are observed within each group.
> In contrast, the representations from MDS show a merging tendency among groups, and each group is latently represented as an isotropic disc area, lacking the subgroup information.
> This is why we say PhyloVAE can provide high-resolution representations compared to distance-based methods.
>
> In the introduction, we did mention that the effectiveness of these distance-based methods (not PhyloVAE) heavily depends on the choice of distance metric and can sometimes exhibit counterintuitive behaviors.
> [Kuhner & Yamato, 2015] give an example of such counterintuitive behaviors where distanced-based methods can provide somewhat discordant results when analyzing trees with different levels of similarity.
> Similar drawbacks of distance-based methods are also discussed in [Kendall & Colijn, 2016].
>
> **W2**: The specific application scenarios described in the manuscript are vague (...)
>
> **Response to W2**:
> Just like the general VAE framework, PhyloVAE can perform generative modeling and representation learning simultaneously. We think this is a nice property of our method.
>
> Although the main purpose of PhyloVAE is indeed for representation learning of phylogenetic topologies, we want to emphasize it is the generative modeling perspective of PhyloVAE (inherited from variational autoencoders) that enables more capacity for high-resolution representations. More specifically, PhyloVAE tries to learn the overall distribution of trees instead of just maintaining pairwise distances between them as typically done in distance-based methods. This generative modeling perspective forces PhyloVAE to learn high-resolution representations to retain more distributional information (see Figure 4 for an example).
> Moreover, the generative modeling nature also allows PhyloVAE to map the latent space back to the tree space, which is impossible for previous distance-based approaches. This can be used in various tasks such as designing novel tree topology proposals from the latent space for more efficient exploration in the tree space.
> For generative modeling, PhyloVAE can be used as an alternative to ARTree, which is much faster to train and sample from, while maintaining the approximation accuracy.
> We will add these discussions to our revision.
>
> **W3**: This paper mentions the limitations of (...)
>
> **Response to W3**:
> Thanks for your question! The main setting for the PhyloVAE model that makes it better at representation learning and visualization than distance-based and density-based methods is that **it combines representation learning and generative modeling in a satisfying and useful way, which is inherited from variational autoencoders (VAE)**. As mentioned in our response to W2, the generative modeling perspective forces PhyloVAE to learn high-resolution representations that retain the distributional information of tree topologies, which contain more information than the pairwise distances used in distance-based methods. Compared to density-based methods, PhyloVAE has a latent space that allows representation learning of tree topologies, while the density-based method only provides estimated densities that contain no structural information of the tree shape distributions.
> We expect models/methods that share the same hybrid settings of latent representation and generative modeling to perform well in representation learning and visualization.
>
> **W4**: More minor issues could be clarified
>
> **Response to W4**:
>
> (a) Different software choices may lead to samples with different accuracies. This is an independent task from representation learning. We only make an effort to show that PhyloVAE can give reliable representations of whatever training samples are given.
>
> (b) There is only a constraint $\sum_i w_i = 1$ without other priori constraints. This is just an empirical distribution, defined by a weighted sum of Diracs.
>
> (c) We select only 5 and 8 because the posterior distribution can be freely designed and **analytically computed** (This is impossible for more than 8 leaves which can have more than $10^6$ different trees). For larger data sets, please see the results in Sections 5.2 and 5.3.
>
> (d) To plot Figure 3 (left), we (i) select $11\times 11$ grid points on the square $(0,1)^2$ (These grid points are the quantiles of the standard Gaussian). (ii) For each grid point $(x,y)$, we compute the tree topology encoding vector $s(\tau)$ through the generative model $p_\theta(s(\tau)|(x,y))$. (iii) Convert $s(\tau)$ to $\tau$ through the reconstruction loop (Algorithm 3). (iv) Plot $\tau$ on the position $(x,y)$.
>
> (e) Yes, they represent the same variable.

---

> > ### Author Response · Authors · 2024-11-22
> > **Response to Reviewer SRrn's concerns - 2**
> >
> > **Q1**: One of the motivations mentioned in the text is: (...)
> >
> > **Response to Q1**:
> > We'd like to first clarify the potential misunderstanding of PhyloVAE.
> > **The map from representations to trees is not a drawback or a limitation, but a major advantage of PhyloVAE.**
> > This map comes from the auto-encoder nature of PhyloVAE and does not exist in distance-based methods (e.g., MDS).
> > In particular, without this map, distance-based methods can only learn vector representations for trees in the training data, and this is why we say they do not form an actual visualization of the tree space.
> > On the other hand, because of this map, PhyloVAE's latent space can be mapped back to the entire tree space (and trees can be encoded to the latent space as well). Therefore, PhyloVAE can provide a complete visualization of the entire tree space.
> > Figure 3 (left) validates the nice continuity of this map.
> > For an example of the practical advantages of this map in PhyloVAE, considering connecting two points in the 2D plane, the curve defines a continuous interpolation between two high-dimensional tree topologies. This may help us to design more effective exploration strategies in phylogenetic MCMC algorithms.
> > These interpretations and applications are impossible for distance-based methods (e.g., MDS).
> >
> > **Q2**: The encoding mechanism in Section 3.2 is clearly written, but there is a potential problem: (...)
> >
> > **Response to Q2**:
> >
> > (a) There might be some potential misunderstanding of PhyloVAE, as clarified below.
> > - The encoding mechanism in Sec 3.2 (including decomposition and reconstruction) is not the generative model.
> > The encoding mechanism, providing a **deterministic mapping** from tree to integer vectors, only requires a tree traversal instead of any dependency on previous actions.
> > - The $Q(\tau|z)=\prod_n Q(e_n|z)$ in your question is relevant to the generative model. In PhyloVAE, all edge decisions $\\{e_n\\}\_n$ are **conditionally independent** given the latent variable $z$. Therefore, as shown by equation (5), one does not have to compute the dependency of $e_n$ on $e_{<n}$. Note that this still allows us to capture complicated dependency given the hierarchical structure of the latent variable model
> > $p_{\theta}(\\tau) = \\int_{\\mathbb{R}^d}p_{\theta}(s(\tau)|z)p(z)\mathrm{d}z,$
> > as demonstrated in VAE for modeling real data distributions such as images and videos.
> >
> > (b) In Sec 3.2, the decomposition process is deterministic, instead of a random distribution. Therefore, there is no error in the decomposition process.
> >
> > **Q3**: The paper claims that PhyloVAE is ‘the first representation learning framework that targets the topology of phylogenetic trees’, (...)
> >
> > **Response to Q3**: Here are our clarifications for this question.
> > - VBPI-GNN only gives a deep model architecture for extracting graph features but does not propose a representation learning method. It should be pointed out that the inference model in PhyloVAE is built on top of VBPI-GNN, which is used as a feature extractor. It is the combination of the VAE framework and VBPI-GNN that allows a complete representation learning procedure of phylogenetic tree topologies.
> > - For the representation learning task, we explore the performance of the widely-used visualization method - MDS plot - as a baseline, shown in Figure 4.
> > - As our model is the first to consider deep models for representation learning on tree topologies, there are no related baselines. Here, we distinguish the visualization methods (e.g., MDS) and deep representation learning methods.

---

> ### Author Response · Authors · 2024-11-22
> **Response to Reviewer SRrn's concerns - 3**
>
> **Q4**: Where is the hyper-parameter experiment, e.g., how to decide the parameters $d$, $K$, $L$, Iterations, and batch size?
>
> **Response to Q4**:
> Thanks for this question. We explain the choice of hyper-parameters as follows:
> - the latent dimension $d=2$ is for ease of visualization; $d=10$ serves as an ablation study with large model capacity.
> - $K=32$ is a just experiential choice, which is fixed across all experiments.
> - The number of layers $L=2$ and iterations simply follows ARTree, the most relevant baseline.
> - Number of iterations and batch size are kept the same as SBN and ARTree.
>
> To more comprehensively evaluate the effect of $d$ and $K$, we provided an additional ablation study in our revision (see Table 3). We also put it here for your convenience. We see that increasing $K$ can generally improve the approximation accuracy, while sometimes a large $d$ may increase the training difficulty and lead to overfitting.
>
> *Table: KL divergence on DS1*
> |KL | K=1 | K=16| K=32| K=64|
> |---|---|---|---|---|
> |d=2|0.1275|0.0308|0.0273|0.0264|
> |d=5|0.0951|0.0182|0.0177|0.0166|
> |d=10|0.0997|0.0230|0.0189|0.0175|
>
> *Table: KL divergence on DS2*
> |KL | K=1 | K=16| K=32| K=64|
> |---|---|---|---|---|
> |d=2|0.0202|0.0097|0.0100|0.0097|
> |d=5|0.0202|0.0103|0.0099|0.0107|
> |d=10|0.0202|0.0107|0.0098|0.0103|
>
> *Table: KL divergence on DS3*
> |KL | K=1 | K=16| K=32| K=64|
> |---|---|---|---|---|
> |d=2|0.0674|0.0482|0.0529|0.0559|
> |d=5|0.1397|0.0461|0.0502|0.0532|
> |d=10|0.0980|0.0453|0.0477|0.0515|
>
> *Table: KL divergence on DS4*
> |KL | K=1 | K=16| K=32| K=64|
> |---|---|---|---|---|
> |d=2|0.1038|0.0646|0.0619|0.0607|
> |d=5|0.0995|0.0470|0.0467|0.0471|
> |d=10|0.1082|0.0470|0.0469|0.0460|

---

> > ### Comment · Reviewer_SRrn · 2024-11-27
> > **Feedback to Authors' Rebuttal**
> >
> > Thanks for the detailed response and additional experiments. I believe my concerns and questions have been almost tackled with several additional comments:
> > As for W2, the authors should further polish the manuscript to clarify the motivation and the targeted problem.
> > Clarification of the difference and novelty of the proposed PhyloVAE might be discussed in the manuscript to overcome potential problems like Q3.
> > After reading reviews from other reviewers, I decided to increase my score to 6. I hope this work will benefit the phylogenetics community.

---

> > > ### Author Response · Authors · 2024-11-28
> > > **Thanks for raising your score!**
> > >
> > > Thank you very much for raising your score! We highly appreciate your suggestions and will include the discussions about W2 and Q3 in our revision.

---

### Author Response · Authors · 2024-11-24
**Global response**

We thank all reviewers for their careful review and constructive feedback.
We have uploaded a revised manuscript with the following major changes.

- We add clarification on the collection of tree topologies that are fed into PhyloVAE in Line 159. These tree topologies are used as training samples whose representations are of interest; it is clearly distinct from [Zhang & Matsen, 2019;2024] where the pre-selected trees are used for support estimation.
- We distinguish PhyloVAE from other previous works that integrated trees with VAEs in Line 348. These works all consider tree-shaped prior distribution or hierarchical latent variable structure and do not consider modeling any graph or tree objects and thus are clearly distinct from PhyloVAE.
- We explain why DS1-8 are good representative data sets in Line 429.
- We add an ablation study about the approximation accuracy as $d$ and $K$ vary in Table 3.
- We add a comparison between tree probability estimates and the ground truth in Figure 13.

---

### Meta-Review · Area_Chair_KPBP · 2024-12-21

**Metareview:**

The paper introduces PhyloVAE, a variational autoencoder (VAE) for representation learning and generative modelling of phylogenetic tree topologies. The reviewers commend the paper for being overall easy to follow, the addressed problems for being highly practical relevant, for the proposed method to be highly innovative, for providing significant efficiency gains over ARTree, and for the experimental results being significant. Concerns raised by the reviewers included a lack of evaluation of generalizability across datasets, marginal accuracy improvements, and questions about the sufficiency of the datasets used. The rebuttal and discussion clarified key aspects, provided additional experimental results, and addressed concerns about computational trade-offs. While some reviewers sought further experiments or discussions (e.g., cross-dataset generalizability, mixtures of variational distributions for further improvement), most agreed the paper offered a meaningful contribution, and I, therefore, recommend accepting the paper.

**Additional Comments On Reviewer Discussion:**

The reburial and discussion points are mentioned in the meta review.

---

### Decision · Program_Chairs · 2025-01-22

Accept (Poster)